# A branched heterochronic pathway directs juvenile-to-adult transition through two LIN-29 isoforms

**Chiara Azzi[1,2†], Florian Aeschimann[1,2†‡], Anca Neagu[1], Helge Großhans[1,2*]**

[1]Friedrich Miescher Institute for Biomedical Research, Basel, Switzerland; [2]University of Basel, Basel, Switzerland

**Abstract** Robust organismal development relies on temporal coordination of disparate physiological processes. In *Caenorhabditis elegans*, the heterochronic pathway controls a timely juvenile-to-adult (J/A) transition. This regulatory cascade of conserved proteins and small RNAs culminates in accumulation of the transcription factor LIN-29, which triggers coordinated execution of transition events. We report that two LIN-29 isoforms fulfill distinct functions. Functional specialization is a consequence of distinct isoform expression patterns, not protein sequence, and we propose that distinct LIN-29 dose sensitivities of the individual J/A transition events help to ensure their temporal ordering. We demonstrate that unique isoform expression patterns are generated by the activities of LIN-41 for *lin-29a*, and of HBL-1 for *lin-29b*, whereas the RNA-binding protein LIN-28 coordinates LIN-29 isoform activity, in part by regulating both *hbl-1* and *lin-41*. Our findings reveal that coordinated transition from juvenile to adult involves branching of a linear pathway to achieve timely control of multiple events.

**\*For correspondence:**
helge.grosshans@fmi.ch

[†]These authors contributed equally to this work

**Present address:** [‡]CSL Behring, Research, CSL Biologics Research Center, Bern, Switzerland

## Introduction

Temporal coordination of diverse events is a hallmark of organismal development. This is illustrated by the juvenile-to-adult (J/A) transition of animals, in mammals also known as puberty. J/A transition involves coordinated morphological changes of sexual organs as well as various other tissues and organs, including skin (*Lee and Houk, 2006*). The molecular mechanisms that control the onset of J/A transition in humans are poorly understood, but have been well studied in the nematode *Caenorhabditis elegans* (*Faunes and Larraín, 2016*). In *C. elegans*, J/A transition is controlled by a cascade of regulators termed the heterochronic pathway, which coordinates somatic cell fate programs (*Ambros and Horvitz, 1984*). Orthologues of heterochronic genes have also been implicated in timing the onset of puberty in mammals including humans (*Abreu et al., 2013*; *Corre et al., 2016*; *Ong et al., 2009*; *Perry et al., 2009*; *Sulem et al., 2009*; *Zhu et al., 2010*) (reviewed in *Faunes and Larraín, 2016*; *Moss and Romer-Seibert, 2014*), indicating an evolutionary conservation of the molecular principles of temporal coordination of J/A transition events.

The *C. elegans* J/A transition has been particularly well studied in the epidermis, where it encompasses four events related to cell fates and molting (*Ambros, 1989*), illustrated in their order of occurrence in *Figure 1A*. First, skin progenitor cells called seam cells, which undergo asymmetric (self-renewal) divisions during larval stages, cease to do so in adult animals. The last seam cell division takes place during the transition from the third (L3) to the last (L4) larval stage. Second, during the mid-L4 stage, seam cells fuse into a syncytium, a state considered terminally differentiated (*Sulston and Horvitz, 1977*). Third, animals generate an adult cuticle, characterized by the presence of adult-specific collagens (*Cox and Hirsh, 1985*) and a microscopically visible structure known as adult alae (*Singh and Sulston, 1978*). Fourth, following shedding of the L4 cuticle, animals stop molting, the process of cuticle synthesis and shedding that happens at the end of each larval stage.

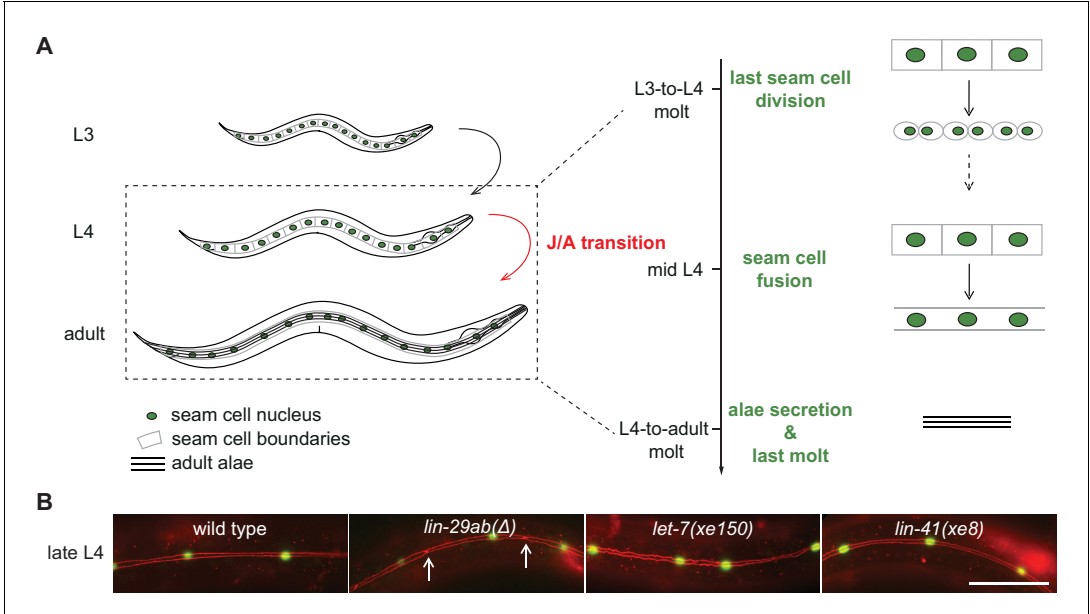

**Figure 1.** Uncoupling of coordinated execution of J/A transition events in *let-7* and *lin-41* mutant animals. (**A**) Schematic representation of juvenile-to-adult (J/A) transition events in the *C. elegans* epidermis: final division of seam cells (square-shaped cells with green nuclei) at the L3-to-L4 molt; seam cell fusion into a syncytium during mid-L4 stage; synthesis of an adult cuticle containing lateral alae (three horizontal bars) at the L4-to-adult molt; and a subsequent exit from the molting cycle. (**B**) Micrographs of late L4-stage animals of indicated genotypes expressing *scm::gfp* (*green*, marking seam cells) and *ajm-1::mCherry* (*red*, marking hypodermal cell junctions). Arrows indicate cell boundaries between unfused cells. Representative of n > 20. Scale bar: 50 μm.

The online version of this article includes the following source data for figure 1:

**Source data 1.** Quantification of unfused seam cell junctions, raw data related to *Figure 1B*.

Genetic screens have identified precocious and retarded mutations (*Ambros and Horvitz, 1984*), which cause animals to exhibit somatically adult features before reaching sexual maturity or retain juvenile somatic features after reaching sexual maturity, respectively. Thus, the identified factors, called heterechronic genes, regulate the initiation of J/A transition events. Among these genes,

*lin-29*, encoding a transcription factor of the EGR/Krüppel family, is considered the downstream-most gene of the heterochronic pathway (*Rougvie and Moss, 2013*). Indeed, LIN-29 accumulates immediately prior to transition to adulthood during the last (L4) larval stage (*Bettinger et al., 1996*).

Current models of the heterochronic pathway depict a simple linear chain of events during the last larval stages that leads to upregulation of LIN-29 (*Faunes and Larraín, 2016*; *Moss and Romer-Seibert, 2014*; *Rougvie and Moss, 2013*). The miRNA *let-7* accumulates during the L3 stage to inhibit synthesis of the RNA binding protein LIN-41 (*Ding and Großhans, 2009*; *Reinhart et al., 2000*). Since LIN-41 translationally represses *lin-29* (*Aeschimann et al., 2017*), its decreased levels during L4 allow for accumulation of LIN-29. As all four epidermal J/A events require LIN-29 (*Ambros and Horvitz, 1984*; *Bettinger et al., 1997*), this pathway architecture can ensure their coordinated execution.

However, multiple lines of evidence challenge this simple linear model. First, LIN-29 occurs in two protein isoforms, LIN-29a and LIN-29b (*Rougvie and Ambros, 1995*), and LIN-41 appears to silence only *lin-29a* but not *lin-29b* (*Aeschimann et al., 2017*). Second, *lin-41(0)* mutant precocious phenotypes, unlike the retarded *lin-29(0)* mutant phenotypes, are only partially penetrant (*Slack et al., 2000*), indicating additional control of LIN-29 beyond repression by LIN-41. Third, *let-7* mutations do not recapitulate all phenotypes of *lin-29(0)* (*Ambros, 1989*), as *let-7* appears dispensable for proper timing of seam cell fusion (*Hunter et al., 2013*).

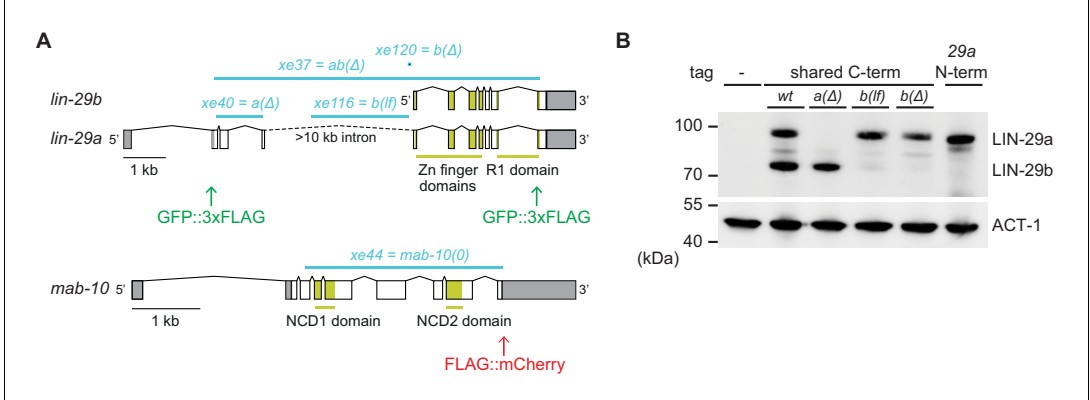

**Figure 2.** Generation of *lin-29a* and *lin-29b* isoform-specific mutants. (**A**) Schematic representation of the *lin-29* and *mab-10* genomic regions. Mutant alleles and endogenously tagged alleles used in this study are indicated. Insertion of a *gfp::3xflag*-encoding sequence at the 5' end specifically tags LIN-29a at its N-terminus, while insertion at the 3' end tags both isoforms at the shared C-terminus. Insertion of this C-terminal tag in a *lin-29(xe40[lin-29a (Δ)])* genetic background yields specific tagging of LIN-29b. Allele numbers refer to modifications in otherwise wild-type backgrounds; numbers for equivalent mutations in the endogenously tagged backgrounds, used for protein detection by microscopy and Western blotting, are provided in the Key resources table. (**B**) Western blot of C-terminally GFP::3xFLAG-tagged endogenous LIN-29a and LIN-29b proteins in the different mutant backgrounds using anti-FLAG antibody. Animals were grown for 36 hr at 25°C to the late L4 stage. Actin-1 is used as loading control.

Here, we address these discrepancies by studying the regulation of the two LIN-29 isoforms and their functions in J/A transition. LIN-29a and LIN-29b share most of their protein sequence with the exception of 142 amino acids at the N-terminus that are unique to LIN-29a (*Figure 2A*). Previous studies have suggested that these isoforms function redundantly: they share a common co-factor, MAB-10 (*Harris and Horvitz, 2011*), they have similar expression patterns, and they are interchangeable in complementation analysis (*Bettinger et al., 1996*; *Bettinger et al., 1997*). However, employing isoform-specific mutations and endogenous protein tagging, we show here that the *lin-29a* and *lin-29b* isoforms differ in function and expression patterns. The most striking functional difference occurs in seam cell fusion, which relies only on LIN-29b, but not on LIN-29a or MAB-10. Moreover, whereas *lin-29a* is regulated by LIN-41, the *lin-29b* isoform is regulated by the transcription factor HBL-1, which results in a distinct spatiotemporal expression of the two isoforms. The expression patterns alone can explain the unique phenotypic consequences of the isoform-specific mutations, whereas the sequence difference in the N-terminal portion of the two isoforms does not contribute to functional differences. Coordination of the activities of LIN-29a and LIN-29b is achieved through LIN-28, an RNA-binding protein that regulates both HBL-1 and *let-7*–LIN-41. Taken together, our findings help to reframe the regulatory logic that enables the heterochronic pathway to coordinate different events into an overall larval-to-adult transition program.

## Results

### Regulation of LIN-29 by *let-7* and LIN-41 is dispensable for triggering seam cell fusion

A model of simple linear control in the heterochronic pathway predicts that *lin-29(0)* mutations cause the same phenotypes as upstream mutations that impair the activation of *lin-29*, such as *let-7(0)*. Indeed, the phenotypes of the two mutant strains overlap extensively (*Ambros and Horvitz, 1984*; *Bettinger et al., 1997*; *Reinhart et al., 2000*). However, they are not identical: Whereas *lin-29(0)* mutant animals fail to execute seam cell fusion (*Bettinger et al., 1997*), this process was reported to occur normally in *let-7(mn112)* mutant animals (*Hunter et al., 2013*). To validate this unexpected observation, we examined a newly created *let-7* null mutant strain, *let-7(xe150)*, which lacks *let-7* expression due to deletion of the promoter (a gift from J. Kracmarova). Additionally, we used the temperature-sensitive *let-7(n2853)* strain (*Reinhart et al., 2000*) at the restrictive temperature, 25°C.

Employing an *ajm-1::mCherry* marker to visualize the seam cells boundaries, we observed that seam cells fused normally in 100% of the *let-7(xe150)* and *let-7(n2853)* mutant animals, confirming that *let-7* is indeed dispensable for seam cell fusion (*Figure 1B*). By contrast, complete loss of *lin-29* in *lin-29(xe37)* null mutant animals (henceforth *lin-29ab(Δ)*; *Aeschimann et al., 2019*) caused failure of seam cell fusion in all animals (*Figure 1B*).

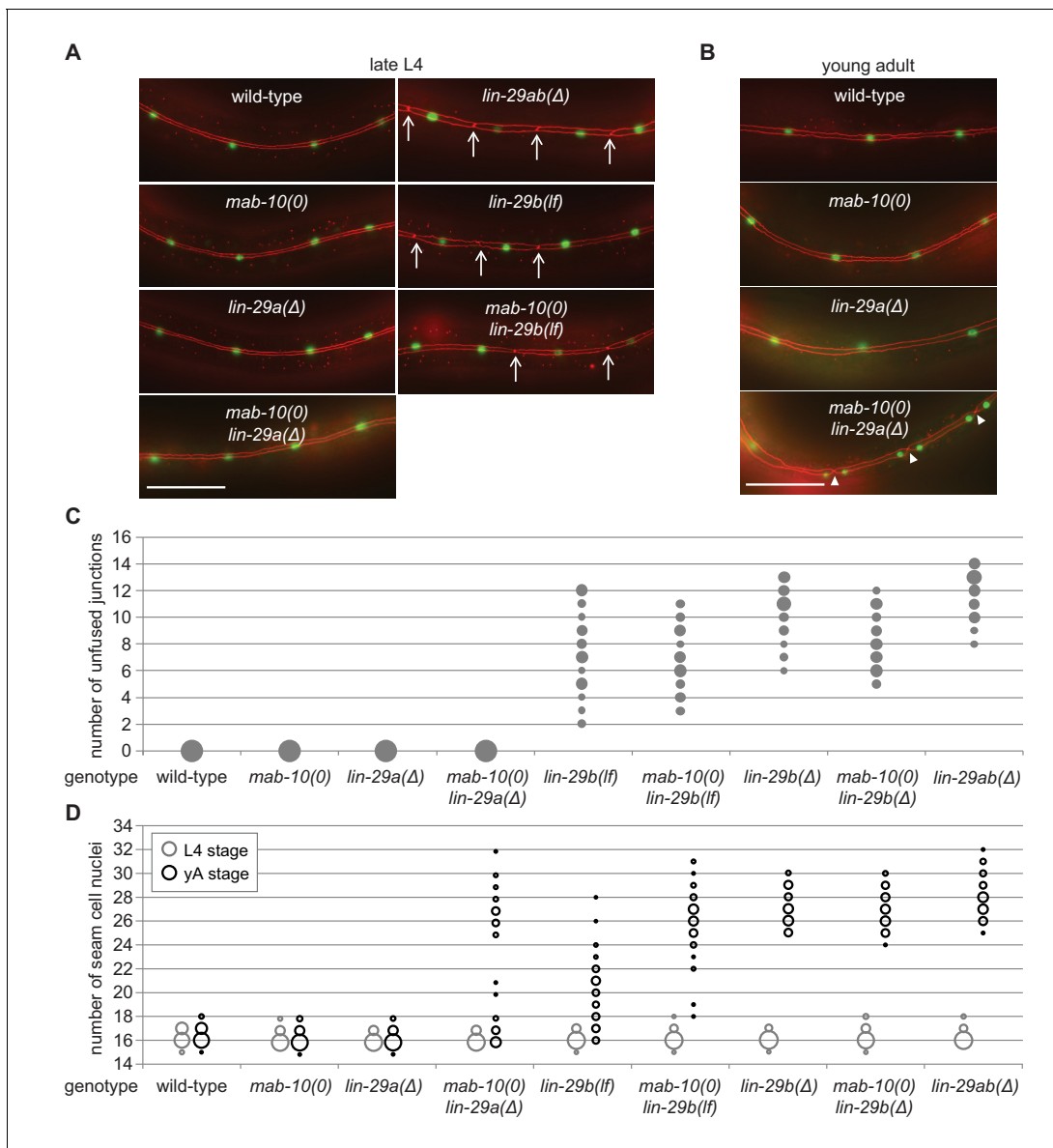

**Figure 3.** LIN-29b has a fundamental role in the regulation of early J/A transition events. (A–B) Micrographs of late L4 stage (A) and young adult (B) animals of indicated genotype expressing *scm::gfp* (*green*, marking seam cells) and *ajm-1::mCherry* (*red*, marking seam cell boundaries). Arrows indicate cell boundaries between unfused seam cells, arrowheads indicate newly formed seam cell boundaries. Scale bars: 50 µm. (C) Quantification of unfused seam cell junctions inL4 larval stage animals of the indicated genetic backgrounds. Areas of bubbles represent the percentage of worms with the respective number of unfused junctions (n > 20 for each genotype). (D) Quantification of seam cell numbers in L4 larval stage and young adult (yA) animals of the indicated genetic backgrounds. Areas of bubbles represent the percentage of worms with the respective number of seam cells (n = 20 for L4, n > 50 for yA worms per genotype).

The online version of this article includes the following source data for figure 3:

**Source data 1.** Quantification of unfused seam cell junctions, raw data related to *Figure 3C*.
**Source data 2.** Quantification of unfused seam cell junctions, raw data related to *Figure 3B*.
**Source data 3.** Quantification of seam cell numbers, raw data related to *Figure 3D*.

Contrasting with the normal seam cell fusion in *let-7* mutant animals, overexpression of *lin-41* had previously been reported to prevent seam cell fusion (*Slack et al., 2000*). However, when we examined *lin-41(xe8[ΔLCS])* mutant animals (*Figure 1B*), which lack the *let-7* complementary sites (LCSs) in the *lin-41* 3'UTR and thus exhibit sustained high LIN-41 levels during the L4 stage (*Aeschimann et al., 2017*; *Ecsedi et al., 2015*), seam cell fusion occurred normally. Hence, the activity of the *let-7*–LIN-41 module cannot account for all the J/A transition events regulated by *lin-29*.

## Seam cell fusion requires LIN-29b but not its co-factor MAB-10 nor LIN-29a

Recently, we showed that *let-7*–LIN-41 regulate *lin-29a* (and *mab-10*), but presumably not *lin-29b* (*Aeschimann et al., 2017*; *Aeschimann et al., 2019*). Accordingly, in *let-7(0)* and *lin-41(ΔLCS)* animals, LIN-29a and MAB-10 levels are expected to remain low in the L4 stage while LIN-29b can accumulate. Hence, we wondered whether LIN-29b would suffice for seam cell fusion and thus explain the phenotypic discrepancy between *let-7(0)* and *lin-29(0)* mutant animals. To test this possibility, we generated two *lin-29b* isoform-specific mutations (Materials and methods). One, where we deleted the putative promoter of *lin-29b*, achieved extensive, but not complete depletion of LIN-29b, while leaving LIN-29a levels unaltered (*Figure 2A,B*). We will refer to it as *lin-29b(lf)*. The other, *lin-29b(Δ)*, where we altered *lin-29b* translation initiation to translate it out of frame, caused a complete loss of LIN-29b, but we cannot exclude a modest depletion of LIN-29a (*Figure 2A,B*). Thus, failure to see a phenotype of interest in the *lin-29b(Δ)* mutant excludes an essential function of the b isoform. Conversely, observation of a phenotype in the *lin-29b(lf)* mutant reveals an essential contribution of the b isoform to this phenotype, although we might under-estimate the extent of this contribution. We engineered each mutation into two different backgrounds, wild-type animals for functional studies, and animals containing a GFP::3xFLAG-tag at the shared C-terminus of the LIN-29 isoforms for expression analysis by Western blotting and imaging (Key resources table).

We used these, and the previously generated *lin-29(xe40)* (henceforth *lin-29a(Δ)*) and *mab-10 (xe44)* (henceforth *mab-10(0)*) mutant strains (*Aeschimann et al., 2019*) to study the individual contributions of LIN-29a, LIN-29b and their co-factor MAB-10 to the different J/A transition events. MAB-10 itself is thought not to directly bind to DNA, but to bind to, and modulate the activity of, both LIN-29 isoforms (*Harris and Horvitz, 2011*). Consequently, in the *lin-29a*- or *lin-29b*-specific single mutants, either of the two LIN-29 isoforms left can still act together with its co-factor MAB-10, while double mutant animals lacking both MAB-10 and one LIN-29 isoform are left with only the other LIN-29 isoform, acting without the co-factor MAB-10.

To determine the individual roles of the LIN-29 isoforms in seam cell fusion, we used the *ajm-1::mCherry* marker to count the number of unfused junctions in the late L4 stage, after the last seam cell division and before the last molt (*Figure 3A,C*). *lin-29a(Δ)* and *mab-10(0)* single mutant animals, as well as *mab-10(0) lin-29a(Δ)* double mutant animals showed unperturbed seam cell fusion. This is consistent with the functional fusion seen in *let-7(0)* and *lin-41(ΔLCS)* animals (*Figure 1B*).

In striking contrast to these findings, we observed penetrant seam cell fusion defects in both *lin-29b(lf)* and *lin-29b(Δ)* single mutant strains, and this phenotype was not enhanced by concomitant loss of *mab-10* expression, that is in *mab-10 lin-29b* double mutant animals (*Figure 3A,C*). These findings support the notion that the *let-7*–LIN-41–LIN-29a/MAB-10 module is dispensable for seam cell fusion. They also reveal an unanticipated specialization of LIN-29 isoforms, with LIN-29b being both necessary and sufficient for wild-type seam cell fusion. We conclude that the two LIN-29 isoforms fulfill distinct and non-redundant functions.

## Fully functional cell cycle exit of seam cells requires both LIN-29 isoforms and their co-factor MAB-10

Prompted by the discovery of a non-redundant function of LIN-29 isoforms in seam cell fusion, we surveyed their individual contributions to other J/A transition events. First, we examined exit of seam cells from the cell cycle. As shown previously (*Aeschimann et al., 2019*; *Ambros and Horvitz, 1984*), *lin-29ab(Δ)* (*lin-29(xe37)*) animals exhibit a fully penetrant phenotype: seam cells do not exit the cell cycle but instead continue to divide so that all animals show at least 25 seam cells instead of

the canonical 16 at the young adult stage (*Figure 3D*). By contrast, seam cells exit the cell cycle normally in *lin-29a(Δ)* and *mab-10(0)* animals. A partially penetrant defect occurs in *mab-10(0) lin-29a(Δ)* double mutant animals (*Figure 3D*), recapitulating the phenotype of *let-7(n2853)* and *lin-41(ΔLCS)* animals (*Aeschimann et al., 2019*). Unlike loss of LIN-29a alone, depletion of LIN-29b alone suffices to permit unscheduled seam cell divisions in young adult animals (*Figure 3D*). The phenotype is more penetrant in *lin-29b(Δ)* than in *lin-29b(lf)* animals. Moreover, loss of *mab-10* enhances the *lin-29b(lf)* but not the *lin-29b(Δ)* mutant phenotype.

Collectively, these data thus confirm an involvement of all three factors, LIN-29a, LIN-29b and MAB-10 in seam cell cell cycle exit. The data also suggest a more prominent role for LIN-29b versus LIN-29a, and a function for MAB-10 in promoting LIN-29 activity in this process.

## Uncoupling of nuclear division and differentiation programs in seam cells

Cell division and terminal differentiation are normally mutually exclusive, tightly coupled events. In fact, exit from the cell division cycle is frequently considered a central aspect of terminal cell differentiation (*Myster and Duronio, 2000*). Hence, we were surprised to find that nuclear divisions continued to occur after seam cells had fused at the L4 stage in *mab-10(0) lin-29a(Δ)* mutant animals (*Figure 3A–D*). We investigated this apparent discrepancy further and found that although in the L4 stage, all seam cells fused in all animals, cell junctions were apparent again in the young adult stage. Specifically, following the unscheduled divisions of seam cell nuclei in these syncytia, pairs of seam cell nuclei were separated from other pairs by cell junctions (*Figure 3B*). Hence, although we have not performed lineaging experiments, we consider it reasonable to assume that these represent newly formed cell boundaries that surround pairs of 'cousins' rather than individual sister cell nuclei. We conclude that exit from the cell cycle is not elicited by cell fusion, and vice versa, that cell fusion does not require permanent cell cycle exit.

## Both LIN-29 isoforms have important but distinct functions in alae formation

To understand LIN-29 isoform function in late J/A transition events, we examined the cuticles of young adults of the different genetic mutant backgrounds by DIC microscopy. An adult cuticle is characterized by alae (*Figure 4A* (I)), ridges along the lateral sides of the whole worm that are secreted by seam cells (*Singh and Sulston, 1978*). Consistent with previous results (*Ambros and Horvitz, 1984*), we observed a complete lack of alae in *lin-29ab(Δ)* young adults (*Figure 4B*).

*lin-29a(Δ)* mutant animals were partially defective in alae formation, exhibiting weak alae structures (*Figure 4A* (II)) that either covered the whole worm ('complete') or at least 50% of the body length ('partial') (*Figure 4B*). The 'partial alae' phenotype appears to reflect a delay in alae synthesis because 4 hr later, all *lin-29a(Δ)* animals had complete, yet still weak, alae (n = 20). Animals depleted of LIN-29b either exhibited only alae patches (*Figure 4A* (III, IV)) or lacked alae entirely (*Figure 4B*), and unlike in the *lin-29a(Δ)* animals, complete alae were not observed even 4 hr later (n = 20). *lin-29b(Δ)* animals had a more penetrant 'no alae' phenotype than *lin-29b(lf)* mutant animals, with almost half of the former animals lacking any detectable alae structure (*Figure 4B*). However, neither *lin-29b* mutation fully recapitulated the *lin-29ab(Δ)* phenotype, and the alae defects were qualitatively distinct in the *lin-29a(Δ)* and the *lin-29b(Δ)* mutant animals. Finally, *mab-10* mutant animals displayed normal, wild-type alae formation (*Harris and Horvitz, 2011*), and absence of MAB-10 did not enhance the phenotypes of any of the *lin-29a* or *lin-29b* mutations (*Figure 4B*).

We conclude that both LIN-29a and LIN-29b, but not MAB-10, are required for wild-type alae formation, and that their functions are partially distinct.

## LIN-29a and LIN-29b function redundantly to regulate the exit from the molting cycle

The last event of the J/A transition that we examined was molting. Using a high-throughput assay (*Meeuse et al., 2020*; *Olmedo et al., 2015*), we counted molts in >20 animals for each genotype (*Figure 5A,B*). We confirmed a previous report (*Ambros and Horvitz, 1984*) that *lin-29* was required for the exit from the molting cycle by observing that all *lin-29ab(Δ)* animals exhibited at least one extra molt, with ~50% exhibiting two extra molts in the time-frame of the experiment. By contrast,

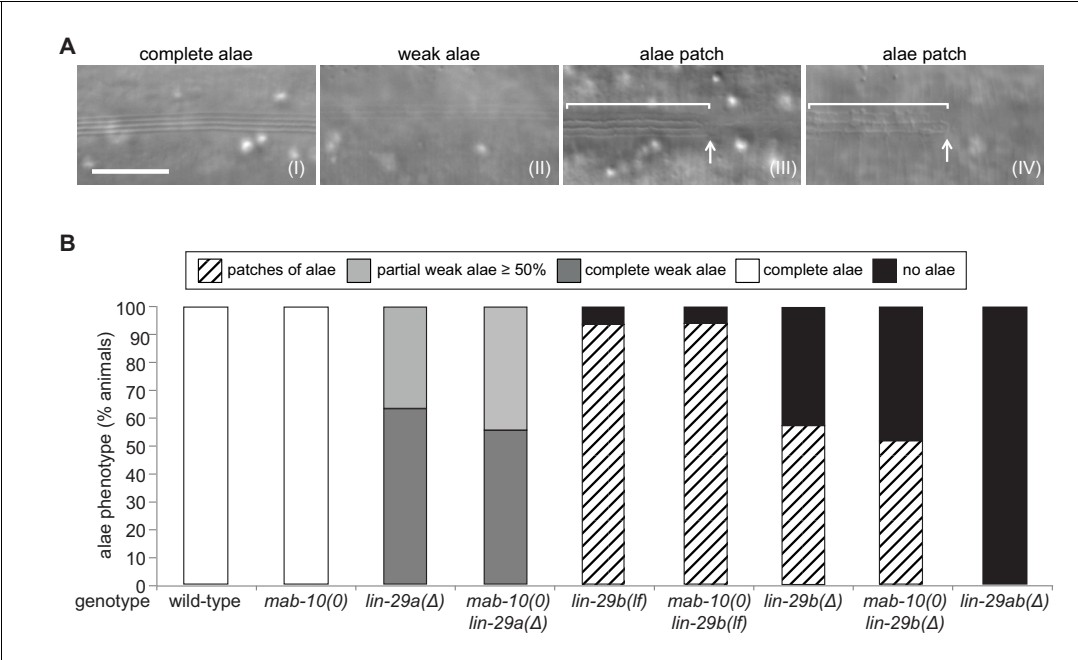

**Figure 4.** LIN-29a and LIN-29b, but not MAB-10, are required for wild-type alae formation. (**A**) Micrographs illustrating categories of alae structures observed on the cuticle of wild-type (I) and mutant (II – IV) young adult animals. The example pictures show (I) wild-type N2, (II) *mab-10(0) lin-29a(Δ)*, (III) *mab-10(0) lin-29b(lf)* and (IV) *mab-10(0) lin-29b(Δ)* animals, respectively. Scale bar: 10 μm. (**B**) Quantification of different alae structures in young adult worms of indicated genotypes (n > 30).

The online version of this article includes the following source data for figure 4:

**Source data 1.** Quantification of alae structures, raw data related to *Figure 4B*.

animals with almost all other single or double mutant combinations, that is *mab-10(0)*, *lin-29a(Δ)*, *lin-29b(lf)*, *lin-29b(Δ)* single mutations and *mab-10(0) lin-29b(lf)* and *mab-10(0) lin-29b(Δ)* double mutations, did not display defects in exiting the molting cycle. The only exception were *mab-10(0) lin-29a(Δ)* animals, of which a small percentage executed one or two extra molts.

We conclude that LIN-29a and LIN-29b have a redundant function in promoting the exit from the molting cycle. Their co-factor MAB-10 has a minor contribution that becomes detectable in a sensitized background.

## Functional differences do not result from difference in molecular sequence

Since our experiments show that LIN-29a and LIN-29b have partially distinct functions (*Figure 6—figure supplement 1*), we asked whether their molecular differences, that is the unique N-terminal extension of 142 amino acids of LIN-29a (*Figure 2A*), are responsible for functional specialization. To address this issue, we genetically engineered the *lin-29* locus such that this N-terminal extension was removed by deleting the *lin-29a*-specific coding exons (exons 2–4) (*Figure 6—figure supplement 2A*). This mutation created a *lin-29a* transcript identical to that of *lin-29b* except for harboring the *lin-29a* 5'UTR and encoding 23 additional N-terminal amino acids. We confirmed by western blotting that this truncated LIN-29a(ΔN) protein had a similar size to LIN-29b and accumulated at roughly wild-type LIN-29a levels (*Figure 6A*). Moreover, the removal of the N-terminus did not affect the spatiotemporal expression pattern of *lin-29a*, and its expression remained under control of LIN-41 (*Figure 6—figure supplement 2C,D*).

We characterized the phenotypes of *lin-29a(ΔN)*, alone or in combination with a *mab-10(0)* allele, for all four J/A transition events. In all assays, deletion of the LIN-29a-specific N-terminus did not compromise LIN-29a function, resulting in a wild-type phenotype for all four J/A transition events

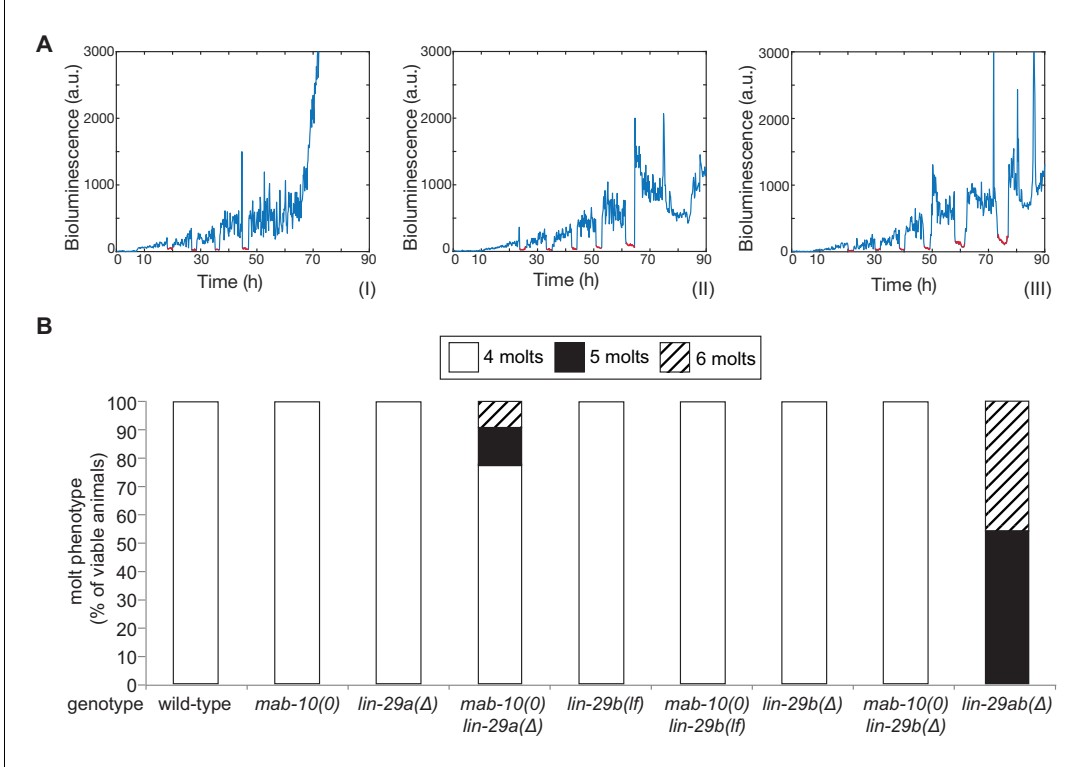

**Figure 5.** The molting cycle is regulated by both LIN-29 isoforms. (**A**) Examples of luciferase assay traces revealing four (I), five (II) or six (III) molts through a drop in luciferase signal (red segment). Examples are from wild-type (I) and *lin-29ab(Δ)* (II-III). (**B**) Quantification of the number of molts in animals of the indicated genotypes (n > 20) based on the assay shown in (**A**). A fraction of *mab-10(0) lin-29a(Δ)* , *lin-29b(Δ)* and *mab-10(0) lin-29b* (Δ) animals die at the J/A transition ; these animals were censored and not included in the quantification.

The online version of this article includes the following source data for figure 5:

**Source data 1.** Quantification of number of molts, raw data related to *Figure 5B*.

(*Figure 6B–D*, *Figure 6—figure supplement 2B*). This data suggests that the distinct functions of LIN-29a and LIN-29b are not mediated by sequence differences, although we cannot formally exclude a contribution of the remaining unique 23 amino acids.

### *lin-29a* and *lin-29b* differ in spatiotemporal expression patterns in the epidermis

Given the apparent absence of differences in molecular function between the two LIN-29 isoforms, we revisited their developmental expression patterns. *lin-29a* and *lin-29b* transcripts were previously reported to exhibit largely similar temporal patterns of accumulation, being both detectable from L1 stage on, with increasing levels during development, peaking at the L4 stage, and decreasing in adulthood (*Rougvie and Ambros, 1995*). Furthermore, a similar spatiotemporal expression pattern was deduced from promoter activity reporter experiments (*Bettinger et al., 1996*). However, transcript quantification and promoter activity measurements do not account for additional layers of post-transcriptional regulation, such as LIN-41-mediated translational repression of *lin-29a* (*Aeschimann et al., 2017*). Previous analysis of LIN-29 protein accumulation by immunofluorescence could not distinguish between the two protein isoforms (*Bettinger et al., 1996*).

To elucidate the temporal expression pattern of *lin-29* isoforms on the protein level, we examined animals carrying a C-terminal GFP::3xFLAG tag that marks both isoforms. Using Western blotting, we could distinguish the two isoforms by their distinct sizes. The levels of the LIN-29b protein agreed with its previously reported pattern of mRNA accumulation, with detectable accumulation throughout larval stages and an increase in levels in the L3 and L4 stages (*Figure 7A*). By contrast,

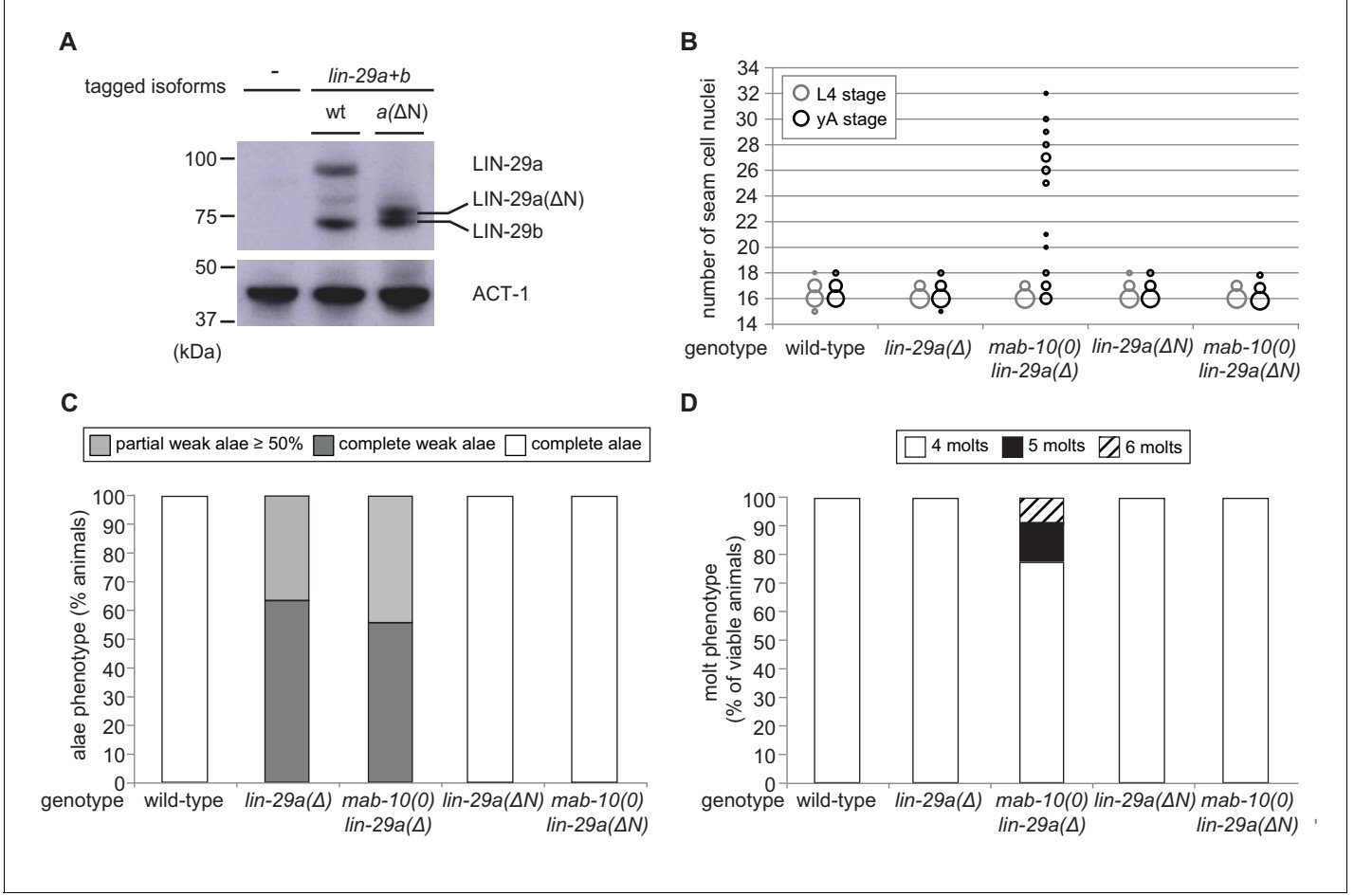

**Figure 6.** The LIN-29a-specific domain is dispensable for the execution of the J/A transition. (A) Western blot of endogenous C-terminally GFP::3xFLAG-tagged LIN-29a and LIN-29b proteins in the *lin-29a(ΔN)* background (HW2408) using an anti-FLAG antibody. Actin-1 is used as loading control. (B) Seam cell number quantification in L4 larval stage and young adult (yA) animals of the indicated genetic backgrounds (n > 25 for L4, n > 25 for yA worms per genotype). The data for *lin-29a(Δ)* and *mab-10(0) lin-29a(Δ)* is re-plotted from *Figure 2* for comparison. (C) Quantification of different alae structures in young adult worms of indicated genotypes (n > 20). The data for *lin-29a(Δ)* and *mab-10(0) lin-29a(Δ)* is re-plotted from *Figure 3* for comparison. (D) Quantification of the number of molts for animals of indicated genotypes (n > 20). The data for *lin-29a(Δ)* and *mab-10(0) lin-29a(Δ)* is re-plotted from *Figure 3* for comparison. In (B – D), *lin-29(ΔN)* is *lin-29(xe200)*.

The online version of this article includes the following source data and figure supplement(s) for figure 6:

**Source data 1.** Quantification of seam cell numbers, raw data related to *Figure 6B*.
**Source data 2.** Quantification of alae structures, raw data related to *Figure 6C*.
**Source data 3.** Quantification of number of molts, raw data related to *Figure 6D*.
**Figure supplement 1.** Summary of the J/A transition phenotypes seen for different permutations of *lin-29a*, *lin-29b*, and *mab-10* mutations.
**Figure supplement 2.** Characterization of *lin-29a(ΔN)* expression and function.
**Figure supplement 2—source data 1.** Quantification of unfused seam cell junctions, raw data related to *Figure 6—figure supplement 2B*.

although the *lin-29a* transcript is detectable in L1 and abundant in L2 (*Rougvie and Ambros, 1995*), LIN-29a protein was undetectable in L1 or L2 stage worms, accumulated weakly in L3 stage, and peaked in L4 stage worms. This is consistent with its post-transcriptional regulation by LIN-41 (*Aeschimann et al., 2017*). We note that in late larval stages and adults, we also detect a third band migrating in between the LIN-29a and LIN-29b bands (*Figure 7A*, asterisk). We did not find evidence for a transcript encoding this intermediate-size protein in available paired-end gene expression data (F. Gypas, personal communication, September 2019) and the molecular nature of this band remains unclear. For the purpose of this study, we considered only the two previously described isoforms (*Rougvie and Ambros, 1995*).

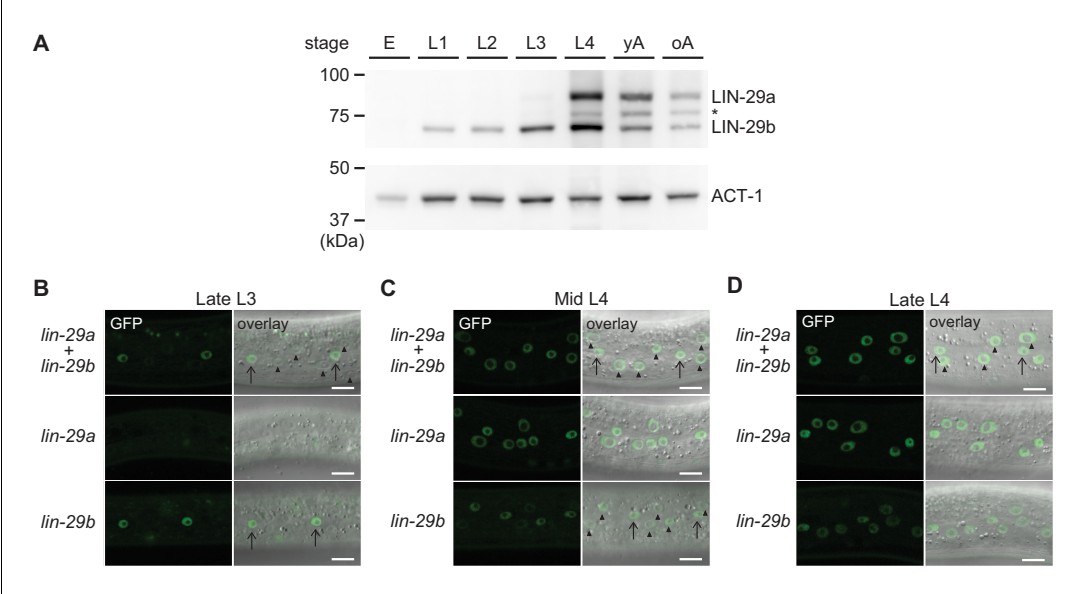

**Figure 7.** Spatial and temporal expression pattern of LIN-29a and LIN-29b. (**A**) Western blot of C-terminally GFP::3xFLAG-tagged endogenous LIN-29a and LIN-29b proteins in different developmental stages using an anti-FLAG antibody. Actin-1 is used as a loading control. The asterisk indicates a band of unclear origin. (**B–D**) Confocal images of endogenously tagged LIN-29 isoforms in the epidermis of animals at the indicated developmental stages, which were confirmed by examination of gonad development. Arrows indicate seam cell, arrowheads hyp7 nuclei. Scale bars: 10 μm.

The online version of this article includes the following figure supplement(s) for figure 7:

**Figure supplement 1.** Expression of *lin-29* isoforms.

To examine the spatiotemporal expression patterns of the two *lin-29* isoforms in vivo, we utilized strains with GFP::3xFLAG tags on the individual, endogenous LIN-29 isoforms. For *lin-29a*, we used a published strain carrying a GFP::3xFLAG tag at the unique N-terminus of LIN-29a (*Aeschimann et al., 2019*). These animals did not display any overt phenotypes. Since we could not create a LIN-29b-specific tag due to lack of isoform-specific sequence, we used a C-terminal fusion that tags both isoforms and mutated the *lin-29a(Δ)* isoform in this background (*Figure 2A*). Hence, we studied *lin-29b* expression in a *lin-29a(Δ)* mutant background. The resulting strain exhibited an increased occurrence of protruding vulva (Pvl) and vulval bursting phenotypes relative to *lin-29a(Δ)* animals without the tag (data not shown) (*Aeschimann et al., 2019*), indicating that the C-terminal tag may partially impair LIN-29 protein function.

Using live imaging of the GFP-tagged proteins, we observed LIN-29 isoform accumulation. In agreement with immunofluorescence-based analysis (*Bettinger et al., 1996*), we detected signal in the epidermis (*Figure 7B–D*) and in other regions such as the head, the tail and the vulva (*Figure 7—figure supplement 1*). In most sites, patterns of the two LIN-29 isoforms appeared to differ. Considering our interest in the role of LIN-29 in the epidermal J/A transition, we focused on the detailed characterization of LIN-29a and LIN-29b accumulation in the skin. LIN-29b was first detectable in lateral seam cell nuclei of late L3 worms (*Figure 7B*). Subsequently, in mid-L4 stage animals, it also accumulated, weakly, in the major hypodermal syncytium hyp7 (*Figure 7C*). By contrast, LIN-29a was not detected in either seam cells or the major hypodermal syncytium hyp7 during the L3 stage (*Figure 7B*). Instead, LIN-29a accumulation began in both cell types in mid-L4 stage worms (*Figure 7C*). Finally, protein levels of both LIN-29a and LIN-29b peaked in lateral seam cells and hyp7 in late L4 stage animals and persisted into adulthood (*Figure 7D*, data not shown).

Taken together, our detailed analysis shows that *lin-29a* and *lin-29b* differ in their spatial as well as temporal expression patterns in the epidermis, likely explaining their distinct function in the four J/A transition events.

## Precocious expression of either *lin-29a* or *lin-29b* can induce seam cell fusion

To formally test if the functional difference between the two *lin-29* isoforms stems from their distinct expression patterns rather than molecular differences, we sought to alter the patterns experimentally. To this end, we expressed either *lin-29a* or *lin-29b* from a single-copy transgene under the control of the epidermal *col-10* promoter, which is active from the first larval stage onwards. To avoid potential toxic effects of temporal misexpression of *lin-29* in early larval stages, we added an auxin-inducible degron (AID) tag (*Zhang et al., 2015*) to the transgenic LIN-29 isoforms and prevented LIN-29 protein accumulation by addition of auxin to maintain the transgenic lines (Materials and methods). We then let animals hatch in the absence of auxin and plated starved, synchronized L1 stage larvae on food to permit LIN-29 protein accumulation, before we examined seam cell fusion, the J/A transition event that relies exclusively on LIN-29b. Strikingly, *col-10*-driven expression of either isoform induced precocious seam cell fusion. Specifically, we observed individual animals with fused seam cells already at 10 hr after plating at 25°C, that is during the L1 stage. At 15 hr after plating, in early L2, all animals had at least partially fused seam cells, with >90% of animals exhibiting complete fusion (*Figure 8*). By contrast, expression of *mab-10* from the *col-10* promoter did not cause any precocious seam cell fusion at this time (*Figure 8*). We conclude that LIN-29 upregulation alone is sufficient to trigger fusion of seam cells, and that upregulation of either LIN-29 isoform suffices. This data further validates that the LIN-29b-specific function observed in seam cell fusion under physiological conditions (*Figure 3A,C*) is not due to sequence differences between the two isoforms, but due to differences in their expression.

## Distinct regulation of *lin-29* isoforms through LIN-41 and HBL-1

We wondered how the distinct expression patterns of *lin-29a* and *lin-29b* are generated. We have previously shown that *lin-29a* is translationally regulated by LIN-41 (*Aeschimann et al., 2017*). However, the regulation of *lin-29b* expression is poorly understood. We noticed that the alae phenotype seen in *lin-29b* mutant adults, small patches of well-formed alae, corresponds to that seen one stage earlier in precocious *lin-41(0)* mutant animals (*Abrahante et al., 2003*). By contrast, the weaker but more continuous alae seen in *lin-29a(Δ)* mutant adults are reminiscent of the appearance of precociously formed alae in HBL-1-depleted L4 stage larvae (*Abrahante et al., 2003*). Hence, we

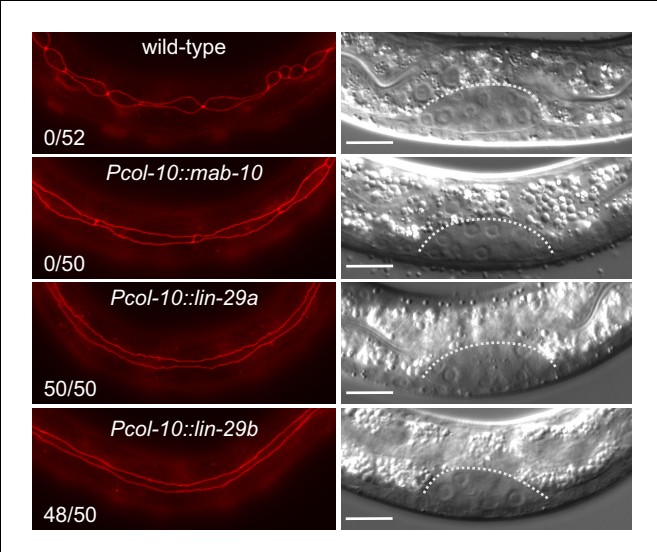

**Figure 8.** Precocious LIN-29 accumulation is sufficient for seam cell fusion in early L2 worms. Micrographs of early L2 animals expressing *ajm-1::mCherry* (*red*, marking hypodermal cell junctions) and the indicated transgenes ('wild-type': no transgene). Animal stage was confirmed by gonad morphology as shown in the DIC pictures. Numbers indicate fractions of animals with complete precocious seam cells fusion. Scale bars: 10 μm.

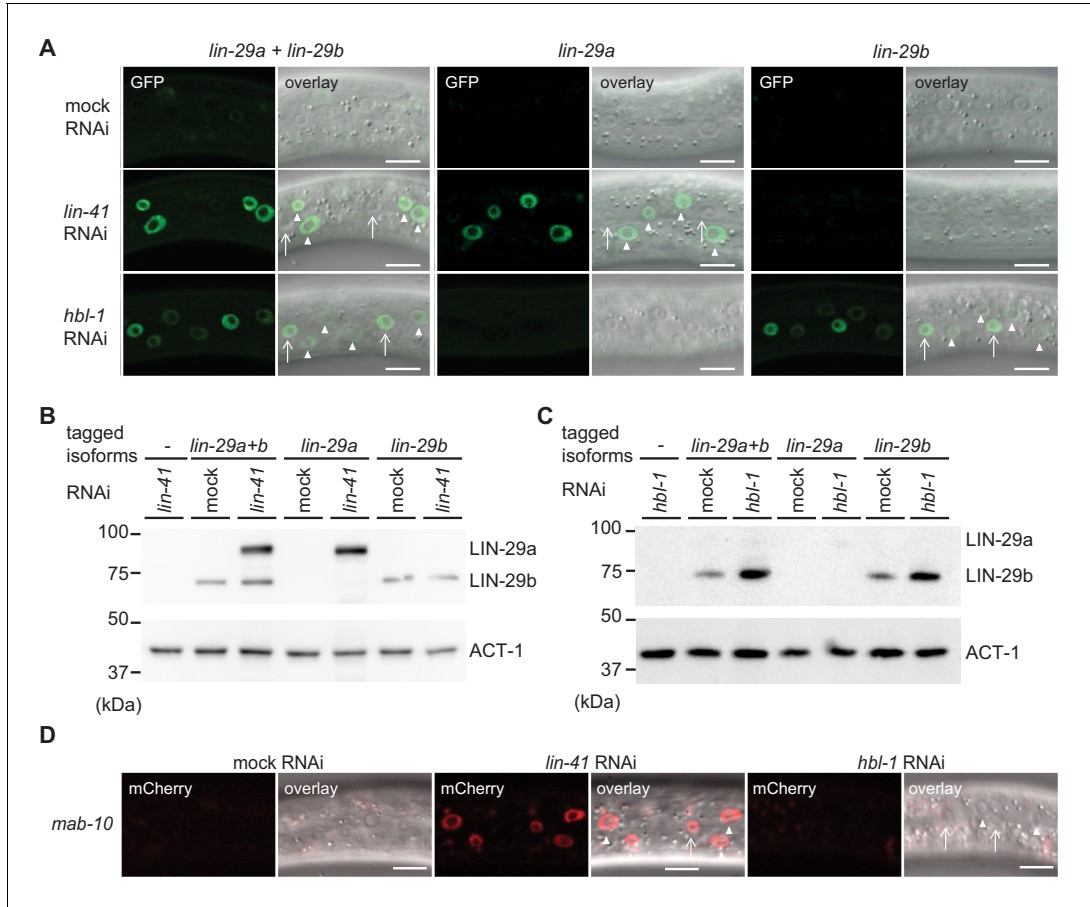

**Figure 9.** *lin-29a* and *lin-29b* expression are specifically regulated by LIN41 and HBL-1, respectively. (**A**) Confocal images of endogenously tagged LIN-29 protein isoforms in the epidermis (strains HW1822, HW1826, HW1835). Animals were grown at 25°C for 20 hr to the early L3 stage on RNAi bacteria as indicated. Arrows indicate seam cell, arrowheads hyp7 nuclei. Scale bars: 10 μm. (**B–C**) Western blot of GFP::3xFLAG-tagged endogenous LIN-29a and LIN-29b proteins using an anti-FLAG antibody. Actin-1 is used as a loading control. Animals were grown for 20 hr at 25°C to the early L3 stage on RNAi bacteria as indicated. (**D**) Confocal images of endogenously tagged MAB-10 protein in the epidermis. Animals were grown at 25°C for 20 hr on *lin-41*, *hbl-1* or mock RNAi bacteria. Arrowheads indicate hyp7 cells, arrows seam cells. Scale bars: 10 μm.

The online version of this article includes the following figure supplement(s) for figure 9:

**Figure supplement 1.** Expression of *lin-29a* on *lin-41* RNAi over time.

**Figure supplement 2.** mRNA levels of *lin-29b* in worms grown on *lin-41* and *hbl-1* RNAi over time.

wondered whether HBL-1, a transcription factor of the Hunchback/Ikaros family (*Abrahante et al., 2003*; *Lin et al., 2003*), and LIN-41 might act on distinct *lin-29* isoforms.

To test this, we examined expression of *lin-29* using the isoform-specific GFP-tags on LIN-29 at the early L3 stage, when neither LIN-29 isoform is expressed in the epidermis under physiological conditions. Consistent with earlier findings (*Aeschimann et al., 2017*), we found that LIN-41 depletion caused precocious accumulation of LIN-29a but not LIN-29b (*Figure 9A,B*). This upregulation was marked in hyp7 by 20 hr after plating but, with a delay, also occurred in seam cells (*Figure 9—figure supplement 1*). By contrast, early L3 stage animals exposed to *hbl-1* RNAi displayed precocious accumulation of LIN-29b but not LIN-29a (*Figure 9A,C*), and this accumulation was more pronounced in the lateral seam cells than in hyp7 (*Figure 9A*). Moreover, it was detectable at the level of the *lin-29b* transcript (*Figure 9—figure supplement 2*). Finally, consistent with earlier evidence (*Aeschimann et al., 2017*), we observed precocious accumulation of a partially functional MAB-10::mCherry protein (*Pereira et al., 2019*) in L3 stage animals depleted of LIN-41 in both seam cells and hyp7, whereas depletion of HBL-1 had no effect (*Figure 9D*).

We conclude that HBL-1 but not LIN-41 regulates LIN-29b accumulation. Conversely, MAB-10 and LIN-29a accumulation are regulated by LIN-41 but not by HBL-1. Whether the regulation of LIN-29b by HBL-1 is direct remains to be determined.

## LIN-28 regulates both *lin-29a* and *lin-29b*

The distinct function and regulation of *lin-29* isoforms that we observed reveals that the heterochronic pathway is not linear but branches into two parallel arms, LIN-41– LIN-29a/MAB-10 and HBL-1– LIN-29b. We wondered where in the pathway this bifurcation occurs. We have previously shown that *lin-41* is the only relevant target of the miRNA *let-7* (*Ecsedi et al., 2015*; *Aeschimann et al., 2017*; *Aeschimann et al., 2019*), placing *let-7* into only one arm of the pathway. The heterochronic gene immediately upstream of *let-7* is *lin-28*, which directly inhibits *let-7* biogenesis (*Lehrbach et al., 2009*; *Van Wynsberghe et al., 2011*). Interestingly, *lin-28* was also reported to positively regulate *hbl-1* (*Vadla et al., 2012*; *Ilbay and Ambros, 2019*), making it a good candidate for being immediately upstream of the pathway's branching point. Hence, we analyzed precocious expression of the LIN-29 protein isoforms in a putative null mutant background of *lin-28*, *lin-28(n719)*, at the early L3 stage. We observed that both isoforms accumulate precociously in *lin-28* mutant animals (*Figure 10*, *Figure 10—figure supplement 1*). Accumulation is initially particularly strong in the seam but over time, and in particular for LIN-29a, also increases in hyp7. Consistent with acting mainly through HBL-1 and LIN-41, respectively, *lin-29b* but not *lin-29a* mRNA levels are increased in *lin-28(n719)* mutant relative to wild-type animals (*Figure 10B,C*). After 22 hr, when LIN-28 is also repressed in wild-type animals, comparable *lin-29b* mRNA levels result. Hence, these data support our hypothesis that heterochronic pathway bifurcation occurs downstream of *lin-28*.

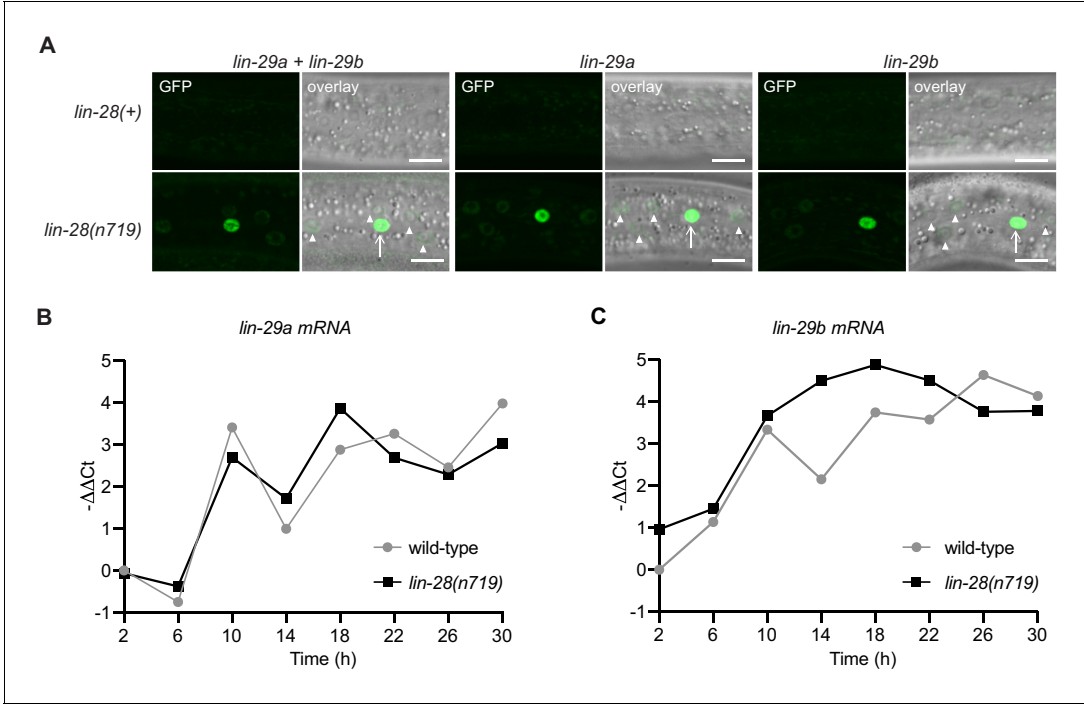

**Figure 10.** LIN-28 coordinates both *lin-29a* and *lin-29b* regulation. (**A**) Confocal images of endogenously tagged LIN-29 protein isoforms in wild-type or *lin-28(n719)* background in the epidermis (strains HW1822, HW1826, HW1835, HW1924, HW1925, HW1926). Animals were grown at 25°C for 20 hr (control strains) and 22 hr (*lin-28(n719)* strains) to reach an equivalent early L3 developmental stage. Arrows indicate seam cell, arrowheads hyp7 nuclei. Scale bars: 10 μm. (**B**) RT-qPCR analysis to measure the -ΔΔCt of *lin-29a* mRNA levels in wild-type and *lin-28(n719)* mutant background (normalized by *act-1* mRNA levels) over time. (**C**) RT-qPCR analysis to measure the -ΔΔCt of *lin-29b* mRNA levels in wild-type and *lin-28(n719)* mutant background (normalized by *act-1* mRNA levels) over time.

The online version of this article includes the following figure supplement(s) for figure 10:

**Figure supplement 1.** Expression of *lin-29a* and *lin-29b* in wild-type and *lin-28* mutant animals over time.

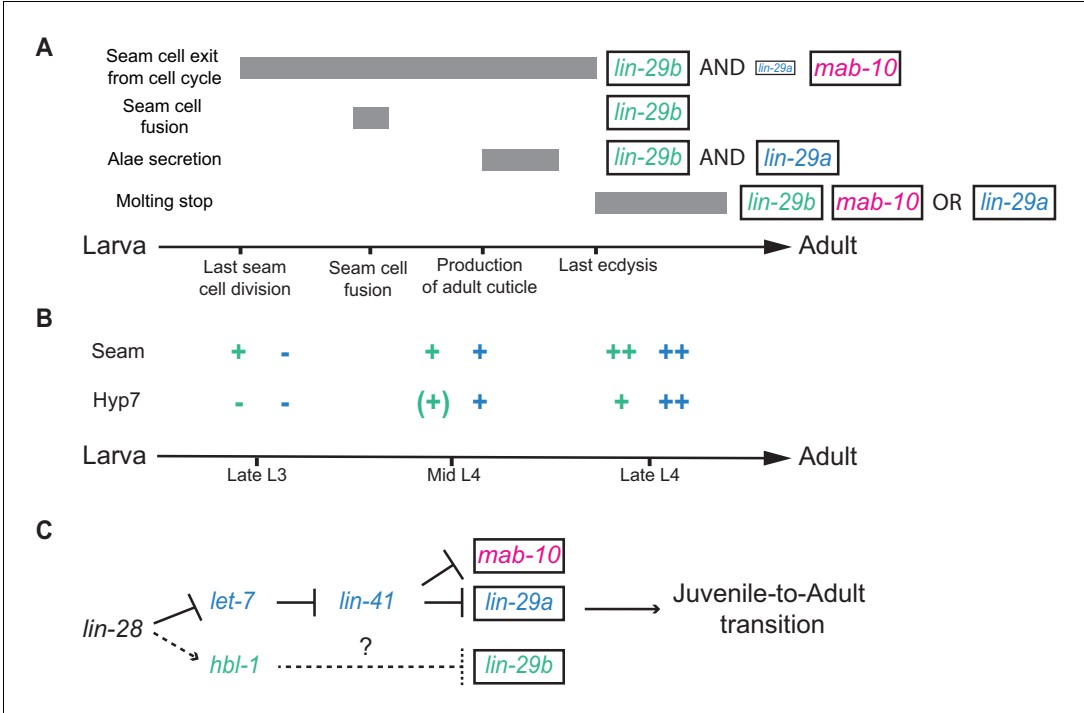

**Figure 11.** Summary. (**A**) Illustration of the contributions of LIN-29 isoforms and the co-factor MAB-10 to the different J/A transition events. The arrow represents developmental time with relevant events indicated. Filled boxes indicate the duration of a specific J/A transition event. The exact time point when seam cells exit the cell cycle is unknown and might occur at any time after the last division at the L3/L4 molt and before the L4/adult molt. Although adults do not molt, it is unknown when exit from the molting cycle occurs. (**B**) Schematic depiction *of lin-29a* (*blue*) and *lin-29b* (*green*) expression patterns in seam cells and hyp7. Relevant developmental stages are indicated on the arrow representing developmental time. (**C**) Revised model of the heterochronic pathway. Two parallel arms of the heterochronic pathway exert their functions through distinct LIN-29 isoforms. Their activities are coordinated throught the upstream function of LIN-28. Dashed arrow indicate indirect regulation which involve *let-7* sisters miRNAs in hyp7 (*Abbott et al., 2005*) and LIN-46 in the seam cells (*Ilbay and Ambros, 2019*; *Ilbay et al., 2019*).

## Discussion

In the four decades since the discovery of the first heterochronic genes (*Ambros and Horvitz, 1984*; *Chalfie et al., 1981*), accumulating genetic and molecular interaction data have facilitated the formulation of ever more refined heterochronic pathway models. A unifying concept of current heterochronic pathway models is that *C. elegans* J/A transition is triggered through a linear chain of events: *let-7* silences LIN-41, and thereby relieves *lin-29* from repression by LIN-41 (*Ambros, 2011*; *Faunes and Larraín, 2016*; *Rougvie and Moss, 2013*). The findings that we have presented here reveal that this concept requires revision. We demonstrate that LIN-29 occurs in two functionally distinct isoforms (*Figure 11A*) and that the functional separation does not depend on differences in protein sequence. Instead, it results from distinct temporal and, possibly, but not directly investigated by us, spatial, expression patterns (*Figure 11B*). Unique expression patterns of the isoforms are the result of isoform-specific regulation: whereas *lin-29a* is under direct translation control by LIN-41 (*Aeschimann et al., 2017*), *lin-29b* is repressed, directly or indirectly, by the transcription factor HBL-1 (*Figure 11C*). Hence, J/A transition relies on two separate arms of the heterochronic pathway rather than a linear chain of events.

The RNA-binding protein LIN-28 regulates both *lin-29* isoforms and thus coordinates their activities. Indeed, we propose that *lin-28* functions as the downstream-most heterochronic gene before branching, consistent with its dual functions of post-transcriptional activation of *hbl-1*

expression (*Vadla et al., 2012*), and inhibition of *let-7* biogenesis (*Lehrbach et al., 2009*; *Van Wynsberghe et al., 2011*), which shields *lin-41* from repression by *let-7*. At the same time, we note that the spatial expression patterns of *lin-29* isoforms upon LIN-28 depletion are predicted by the results of HBL-1 depletion but not entirely by those of LIN-41 depletion. Specifically, whereas upon *lin-41* RNAi,

LIN-29a accumulation in hyp7 precedes that in the seam, the converse is true in *lin-28* mutant animals. While the different kinetics and extents of LIN-41 depletion resulting from the two distinct manipulations may account for some of these differences, the results also suggest that additional, as yet unidentified, regulatory mechanisms exist on which LIN-28 may impact. Indeed, tissue specificity of heterochronic regulation may deserve further attention. In particular, distinct mechanisms appear to repress *hbl-1* in hyp7 versus seam cells: Whereas in hyp7, repression relies on post-transcriptional silencing of the *hbl-1* mRNA by the *let-7* sister miRNAs (miR-48, miR-84 and miR-241) (*Abbott et al., 2005*), in seam cells, LIN-46 sequesters HBL-1 in the cytoplasm after LIN-46 has itself been released from posttranscriptional repression by LIN-28 (*Ilbay and Ambros, 2019*; *Ilbay et al., 2019*).

Our findings and the revised heterochronic pathway model that we propose (*Figure 11C*) finally provide a mechanistic explanation for the previously reported redundancies between *lin-41* and *hbl-1* (*Abrahante et al., 2003*). However, they appear contradicted by other claims and findings from the published literature. First, the functional redundancy of LIN-29 isoforms inferred previously (*Bettinger et al., 1997*) disagrees with our finding of LIN-29 isoform specialization. Second, *lin-41* was described to affect seam cell fusion in both loss-of-function (*Großhans et al., 2005*) and overexpression experiments (*Slack et al., 2000*). This contrasts with a previous report that the *lin-41* regulator *let-7* was dispensable for seam cell fusion (*Hunter et al., 2013*) and, more explicitly, our conclusion that seam cell fusion relies on HBL-1–LIN-29b but not on *let-7*–LIN-41–LIN-29a. Finally, the new model, just like previous ones, fails to accommodate the apparent dual and antagonistic functions of HBL-1, namely suppression of adult cell fates in larvae, and suppression of larval cell fates in adults (*Lin et al., 2003*).

However, closer examination reveals that the published data are fully compatible with the new model. First, the conclusion that *lin-29b* is functionally equivalent to *lin-29a* was based on complementation of *lin-29(0)* mutant heterochronic phenotypes by a *lin-29b* transgene. In these experiments, *lin-29b* was expressed from a multicopy array (*Bettinger et al., 1997*) and thus likely overexpressed. Hence, rather than contradicting our model, these data provide further support for our conclusion that it is *lin-29* isoform expression, not protein sequence, that determines functional distinctions.

Second, although *lin-29a* is dispensable for seam cell fusion in a wild-type context, our experiments (*Figure 8*) clearly demonstrate that, if expressed sufficiently early, *lin-29a* can induce seam cell fusion. This may precisely be the scenario in LIN-41-depleted animals, where LIN-29a (and MAB-10) accumulate precociously (*Figure 9A,B,D*, *Figure 9—figure supplement 1* and *Aeschimann et al., 2017*), thereby causing partial precocious seam cell fusion (*Großhans et al., 2005*).

Seam cell fusion failure in *lin-41* overexpression animals was inferred from the presence of seam cell junctions at the L4/adult molt, rather than from direct observation of syncytium formation in mid-L4 (*Slack et al., 2000*). Therefore, the phenotype seems readily explained by reappearance of junctions following a failure of syncytial nuclei to exit the division cycle. This scenario is supported by our finding that although seam cells fuse in a wild-type manner in both *lin-41* overexpressing *lin-41(ΔLCS)* animals (*Figure 1B*) and *mab-10(0) lin-29a(Δ)* mutant animals (*Figure 3A,C*) in the L4 stage, *mab-10(0) lin-29a(Δ)* mutant adults exhibit de novo generated seam cell junctions (*Figure 3B*).

Third, we propose that a similar scenario of intrasyncytial cell divisions following seam cell fusion also explains the perplexing observation that *hbl-1* mutant adults exhibit seam cell junctions and a greater than wild-type number of seam cell nuclei (*Lin et al., 2003*), which, at the time, was interpreted as a retarded heterochronic phenotype. However, as we find that only LIN-29b accumulates precociously in HBL-1-depleted animals, we would expect them to undergo only a partial J/A transition at the L3 stage. Specifically, although seam cells fuse into a syncytium during L3, syncytial nuclei divide again in L4 and remain trapped in the seam (*Abrahante et al., 2003*). Hence, we propose that *hbl-1* mutant animals exhibit a purely

precocious phenotype and that extra nuclei and seam cell junctions observed in *hbl-1* mutant adults are a consequence of an incomplete precocious J/A transition at the L3/L4 stage rather than reflection of a distinct, larval-fate suppressing function of HBL-1 in adults. Taken together, we find that the new model parsimoniously explains previously published and the present new data, and provides explanations for previously unaccounted phenotypes.

Ours and previous work (*Abrahante et al., 2003*; *Lin et al., 2003*) reveal a striking loss of coordination of J/A transition events in both HBL-1-depleted larvae and *mab-10(0) lin-29a(Δ)* double mutant adults. Specifically, in both cases, seam cells differentiate, as evidenced by fusion and alae formation, but fail to arrest the cell division cycle. This reveals that in seam cells, cell cycle arrest is not necessary for differentiation; that is proliferation and differentiation are not simply antagonistic processes. This finding highlights a need for factors that coordinate the activities of the two arms of the heterochronic pathway, and thus overall J/A transition. We have identified LIN-28 as a relevant factor, but others may exist. Nonetheless, coordination of J/A transition events through upstream activities of the heterochronic pathway rather than the previously proposed 'terminal' *let-7*–LIN-41–LIN-29 axis would seem to entail a surprising lack of robustness, making the pathway vulnerable to perturbations that can uncouple individual J/A transition events. Hence, we will be curious to learn whether this architecture is owed to other, unknown constraints or evolutionary history, or whether it has a particular benefit for the animal. At any rate, we speculate that additional layers of regulation will facilitate coordinated execution of the J/A transition.

Accumulation of LIN-29 has long been considered a key event for triggering J/A transition. We support this key function of LIN-29 in J/A transition by showing that LIN-29 accumulation alone is sufficient to trigger seam cell fusion as early as the L1 stage. While opposite temporal transformation phenotypes for loss-of-function and gain-of-function alleles are considered a hallmark of core heterochronic genes (*Moss and Romer-Seibert, 2014*), rather surprisingly, this had not previously been shown for LIN-29, and it had indeed been argued that LIN-29 accumulation was insufficient to trigger precocious J/A transition (*Slack et al., 2000*). However, our results are further supported by recent molecular evidence showing that *lin-29* is both necessary and sufficient for expression of adult collagens (*Abete-Luzi and Eisenmann, 2018*) and, in parallel to the present study, that precocious expression of *lin-29* in late L2 suffices for seam cell fusion in L3 (*Abete-Luzi et al., 2020*).

We propose that, in the unperturbed system, the temporal order of the individual epidermal J/A transition events is driven by differences in their LIN-29 dose sensitivity and the spatiotemporal differences in *lin-29* isoform expression (*Figure 11A and B*). Specifically, we hypothesize that seam cell fusion rely on LIN-29 accumulation in only the seam, explaining its early occurrence and dependence on only LIN-29b, which accumulates early in this tissue. By contrast, accumulation of LIN-29b alone appears insufficient to drive cessation of seam cell division cycles. Although the last seam cell division normally happens at the L3-to-L4 molt, it is unclear when these cells exit the cell cycle (*Figure 11A*). In fact, our data suggest the possibility of a gradual effect where LIN-29a and LIN-29b can, individually, and more extensively jointly, delay division, that is slow down the cell cycle, but where terminating it requires full LIN-29 activity, and thus MAB-10. Consistent with this notion, exit from the mitotic cycle is the only epidermal J/A transition event for which we observed a defect in *mab-10(0)* single mutant animals, but the additional divisions occurred much later than in the other mutant combinations (data not shown). We note that others reported also extra molts in, mostly older and male, *mab-10* mutant animals (*Harris and Horvitz, 2011*), which we did not investigate. An interesting possibility is that, akin to seam cell division, in the absence of MAB-10, LIN-29a and/or LIN-29b may suffice to delay the occurrence of the next molt rather than to prevent it entirely, and in all animals.

Finally, although both isoforms (but not MAB-10) are required for wild-type alae formation, their individual loss causes qualitatively different defects. We speculate that this may reflect an involvement of hyp7, in addition to seam cells, in alae formation, either directly or through an effect on seam cells that remains to be elucidated. The fact that MAB-10 function is not required for wild-type alae formation, but enhances precocious alae formation in *lin-41* mutant animals (*Harris and Horvitz, 2011*), is then in agreement with the idea that it has a generic function in boosting LIN-29 activity, rather than a specific effect on a subset of LIN-29-regulated targets required for only certain processes. We propose that genome-wide identification of the targets of the *lin-29* isoforms and *mab-10* will allow further testing of this notion in the future. Tissue-specific identification of targets in particular would provide a more detailed understanding of when, where and how different J/A

events are triggered. Already, the findings presented in this study have helped to revise our understanding of the fundamental regulatory architecture that temporally controls events occurring during the transition from a juvenile to an adult animal.

# Materials and methods

## Key resources table

| Reagent type (species) or resource | Designation | Source | Identifiers | Additional information |
|---|---|---|---|---|
| Strain, strain background *Caenorhabditis elegans* | N2 | CGC | N2 | Genotype: Wild-type Figure: all |
| Strain, strain background *Caenorhabditis elegans* | SX346 | (*Lehrbach et al., 2009*) PMID:19713957 | | Genotype: *unc119(e2598) III; wIs51[scm::gfp] V; lin-15(n765) X; mjIs15[ajm-1::mCherry]* Figure: 1B, 3A, 3B, 3C, 3D, 6B, 8, 6S1B |
| Strain, strain background *Caenorhabditis elegans* | HW1387 | (*Aeschimann et al., 2019*) PMID:30910805 | | Genotype: *wIs51[scm::gfp] V; mjIs15[ajm-1::mCherry]; let-7(n2853) X* Figure:1B |
| Strain, strain background *Caenorhabditis elegans* | HW2960 | This study | | Genotype: *let-7(xe150(ΔPlet-7)) X; xeEx365 [Plet-7::let-7::sl1_operon_gfp, unc-119 (+); Prab-3::mCherry; Pmyo-2::mCherry; Pmyo-3::mCherry]; wIs51 [scm::gfp] V; mjIs15[ajm-1::mCherry]* Figure:1B |
| Strain, strain background *Caenorhabditis elegans* | HW1865 | (*Aeschimann et al., 2019*) PMID:30910805 | | Genotype: *lin-41(xe8)/lin-41 (bch28[Peft-3::gfp::h2b::tbb-2 3'UTR) xe70) I; wIs51[scm::gfp] V; lin-15(n765) X; mjIs15[ajm-1::mCherry]* Figure:1B |
| Strain, strain background *Caenorhabditis elegans* | HW1822 | (*Aeschimann et al., 2019*) PMID:30910805 | | Genotype: *lin-29(xe61 [lin-29::gfp::3xflag]) II* Figure: 2B, 6A, 7A, 7B, 7C, 7D, 9A, 9B, 9C, 10A, 6S1C, 6S1D, 7S1A, 7S1B, 7S1C, 9S1A, 9S1B, 10S1A, 10S1B, 10S1C |
| Strain, strain background *Caenorhabditis elegans* | HW1826 | (*Aeschimann et al., 2019*) PMID:30910805 | | Genotype: *lin-29(xe63 [gfp::3xflag::lin-29a]) II* Figure: 2B, 6A, 7A, 7B, 7C, 7D, 9A, 9B, 9C, 10A, 7S1A, 7S1B, 7S1C, 9S1A, 9S1B, 10S1A, 10S1B, 10S1C |
| Strain, strain background *Caenorhabditis elegans* | HW1835 | (*Pereira et al., 2019*) PMID:30599092 | | Genotype: *lin-29(xe40 xe65 [lin-29b::gfp::3xflag]) II* Figure: 2B, 6A, 7A, 7B, 7C, 7D, 9A, 9B, 9C, 10A, 7S1A, 7S1B, 7S1C, 10S1A, 10S1B, 10S1C |
| Strain, strain background *Caenorhabditis elegans* | HW2335 | This study | | Genotype: *lin-29(xe61 xe114 [lin-29::gfp::3xflag])/mnC1 II* Figure: 2B |
| Strain, strain background *Caenorhabditis elegans* | HW2353 | This study | | Genotype: *lin-29(xe61 xe121 [lin-29::gfp::3xflag])/mnC1 II* Figure: 2B |

*Continued on next page*

*Continued*

| Reagent type (species) or resource | Designation | Source | Identifiers | Additional information |
|---|---|---|---|---|
| Strain, strain background *Caenorhabditis elegans* | HW1861 | (*Aeschimann et al., 2019*) PMID:30910805 | | Genotype: *lin-29a(xe40)* II; *wIs51[scm::gfp]* V; *mjIs15[ajm-1::mCherry]* Figure: 3A, 3B, 3C, 3D, 4A, 4B, 6B |
| Strain, strain background *Caenorhabditis elegans* | HW1862 | (*Aeschimann et al., 2019*) PMID:30910805 | | Genotype: *mab-10(xe44)* II; *wIs51[scm::gfp]* V; *mjIs15[ajm-1::mCherry]* Figure: 3A, 3B, 3C, 3D, 4A, 4B |
| Strain, strain background *Caenorhabditis elegans* | HW1860 | This study | | Genotype: *lin-29 (xe37)/mnC1 II*; *wIs51[scm::gfp]* V; *lin-15 (n765)* X; *mjIs15[ajm-1::mCherry]* Figure: 1B, 3A, 3C, 3D, 4A, 4B |
| Strain, strain background *Caenorhabditis elegans* | HW1864 | This study | | Genotype: *mab-10(xe44) lin-29(xe40)/mnC1*; *wIs51[scm::gfp]* V; *lin-15(n765)* X; *mjIs15 [ajm-1::mCherry]* Figure: 3A, 3B, 3C, 3D, 4A, 4B, 6B |
| Strain, strain background *Caenorhabditis elegans* | HW2469 | This study | | Genotype: *lin-29(xe116)* II; *wIs51[scm::gfp]* V; *mjIs15[ajm-1::mCherry]* Figure: 3A, 3C, 3D, 4A, 4B |
| Strain, strain background *Caenorhabditis elegans* | HW2471 | This study | | Genotype: *lin-29(xe120)* II; *wIs51[scm::gfp]* V; *mjIs15[ajm-1::mCherry]* Figure: 3A, 3C, 3D, 4A, 4B |
| Strain, strain background *Caenorhabditis elegans* | HW2480 | This study | | Genotype: *mab-10(xe44) lin-29(xe116)/mnC1* II; *wIs51[scm::gfp]* V; *mjIs15 [ajm-1::mCherry]* Figure: 3A, 3C, 3D, 4A, 4B |
| Strain, strain background *Caenorhabditis elegans* | HW2481 | This study | | Genotype: *mab-10(xe44) lin-29(xe120)/mnC1* II; *wIs51 [scm::gfp]* V; *mjIs15 [ajm-1::mCherry]* Figure: 3A, 3C, 3D, 4A, 4B |
| Strain, strain background *Caenorhabditis elegans* | HW1949 | This study | | Genotype: *xeSi301 [Peft-3::luc::gfp::unc-54 3' UTR, unc-119(+)]* III Figure: 5B, 6D |
| Strain, strain background *Caenorhabditis elegans* | HW2085 | This study | | Genotype: *lin-29(xe37)*; *xeSi301[Peft-3::luc::gfp::unc-54 3'UTR, unc-119(+)]* III Figure: 5B |
| Strain, strain background *Caenorhabditis elegans* | HW2086 | This study | | Genotype: *lin-29(xe40)*; *xeSi301[Peft-3::luc::gfp::unc-54 3'UTR, unc-119(+)]* III Figure: 5B, 6D |
| Strain, strain background *Caenorhabditis elegans* | HW2087 | This study | | Genotype: *mab-10(xe44)*; *xeSi301[Peft-3::luc::gfp::unc-54 3'UTR, unc-119(+)]* III Figure: 5B |

*Continued on next page*

*Continued*

| Reagent type (species) or resource | Designation | Source | Identifiers | Additional information |
|---|---|---|---|---|
| Strain, strain background *Caenorhabditis elegans* | HW2137 | This study | | Genotype: *mab-10(xe44) lin-29(xe40) II; xeSi301[Peft-3::luc::gfp::unc-54 3' UTR, unc-119(+)] III* Figure: 5B, 6D |
| Strain, strain background *Caenorhabditis elegans* | HW2377 | This study | | Genotype: *lin-29(xe116) II; xeSi301[Peft-3::luc::gfp::unc-54 3'UTR, unc-119(+)] III* Figure: 5B |
| Strain, strain background *Caenorhabditis elegans* | HW2378 | This study | | Genotype: *mab-10(xe44) lin-29(xe116)/mnC1 II; xeSi301[Peft-3::luc::gfp::unc-54 3'UTR, unc-119(+)] III* Figure: 5B |
| Strain, strain background *Caenorhabditis elegans* | HW2410 | This study | | Genotype: *lin-29(xe120) II; xeSi301[Peft-3::luc::gfp::unc-54 3'UTR, unc-119(+)] III* Figure: 5B |
| Strain, strain background *Caenorhabditis elegans* | HW2504 | This study | | Genotype: *mab-10(xe44) lin-29(xe120)/mnC1 II; xeSi301 [Peft-3::luc::gfp::unc-54 3'UTR, unc-119(+)] III* Figure: 5B |
| Strain, strain background *Caenorhabditis elegans* | HW2408 | This study | | Genotype: *lin-29(xe61 xe133 [lin-29::gfp::3xflag]) II* Figure: 6A, 6S1C, 6S1D |
| Strain, strain background *Caenorhabditis elegans* | HW2819 | This study | | Genotype: *lin-29(xe200) II; wIs51[scm::gfp] V; mjIs15 [ajm-1::mCherry]* Figure: 6B, 6S1B |
| Strain, strain background *Caenorhabditis elegans* | HW2831 | This study | | Genotype: *mab-10(xe44)lin-29(xe200) II; wIs51[scm::gfp] V; mjIs15[ajm-1::mCherry]* Figure: 6B, 6S1B |
| Strain, strain background *Caenorhabditis elegans* | HW2764 | This study | | Genotype: *lin-29 [(xe200) lin-29a exon 2–4 deletion] II; xeSi301 [Peft-3::luc::gfp::unc-54 3'UTR, unc-119(+)] III* Figure: 6D |
| Strain, strain background *Caenorhabditis elegans* | HW2859 | This study | | Genotype: *mab-10(xe44) lin-29 [(xe200) lin-29a exon 2–4 deletion] II; xeSi301[Peft-3::luc::gfp::unc-54 3'UTR, unc-119(+)] III* Figure: 6D |
| Strain, strain background *Caenorhabditis elegans* | IFM155 | This study | | Genotype: *unc-119(ed3) bchSi39[Peft-3::TIR1::tbb-2 3'UTR, Punc-119::degron-unc-119::unc-119 3'UTR] III* Figure: 8 |
| Strain, strain background *Caenorhabditis elegans* | HW2349 | This study | | Genotype: *ttTi5605 II (EG6699); bchSi39[Peft-3::TIR1::tbb-2 3'UTR, Punc-119::degron::unc-119::unc-119 3'UTR] unc-119(ed3) III* Figure: 8 |

*Continued on next page*

*Continued*

| Reagent type (species) or resource | Designation | Source | Identifiers | Additional information |
|---|---|---|---|---|
| Strain, strain background *Caenorhabditis elegans* | HW2350 | This study | | Genotype: *bchSi39[Peft-3::TIR1::tbb-2 3'UTR, Punc-119::degron-unc-119::unc-119 3'UTR] unc-119(ed3) III; oxTi365 V (EG8082)*<br>Figure: 8 |
| Strain, strain background *Caenorhabditis elegans* | HW2505 | This study | | Genotype: *xeSi422[Pcol-10::flag-ha-degron::lin-29a::lin-29 3'UTR::operon linker (SL2) with gfp-h2b::tbb-2 3'UTR] II; bchSi39[Peft-3::TIR1::tbb-2 3'UTR, Punc-119::degron-unc-119::unc-119 3'UTR] III; mjIs15 [ajm-1::mCherry]\**<br>Figure: 8 |
| Strain, strain background *Caenorhabditis elegans* | HW2506 | This study | | Genotype: *xeSi424[Pcol-10::flag-ha-degron::mab-10::mab-10 3'UTR::operon linker (SL2) with gfp-h2b::tbb-2 3'UTR] V; bchSi39[Peft-3::TIR1::tbb-2 3'UTR, Punc-119::degron-unc-119:: unc-119 3'UTR] III; mjIs15 [ajm-1::mCherry]\**<br>Figure: 8 |
| Strain, strain background *Caenorhabditis elegans* | HW2604 | This study | | Genotype: *xeSi423 [Pcol-10::flag-ha-degron::lin-29b::lin-29 3'UTR::operon linker (SL2) with gfp-h2b::tbb-2 3'UTR] II; bchSi39[Peft-3::TIR1::tbb-2 3'UTR, Punc-119::degron-unc-119::unc-119 3'UTR] III; mjIs15[ajm-1::mCherry]\**<br>Figure: 8 |
| Strain, strain background *Caenorhabditis elegans* | HW2047 | (*Pereira et al., 2019*) PMID:30599092 | | Genotype: *mab-10 (xe75[mab-10::flag::mCherry])*<br>Figure: 9D |
| Strain, strain background *Caenorhabditis elegans* | HW1924 | This study | | Genotype: *lin-28(n719) I; lin-29(xe61[lin-29::gfp::3xflag]) II*<br>Figure: F10A |
| Strain, strain background *Caenorhabditis elegans* | HW1925 | This study | | Genotype: *lin-28(n719) I; lin-29(xe63[gfp::3xflag::lin-29a]) II*<br>Figure: 10A |
| Strain, strain background *Caenorhabditis elegans* | HW1926 | This study | | Genotype:*lin-28(n719) I; lin-29(xe65 [lin-29b::gfp:: 3xflag; lin-29(xe40)]) II*<br>Figure: 10A |
| Strain, strain background *Caenorhabditis elegans* | MT1524 | (*Moss et al., 1997*) PMID:9054503 | | Genotype: *lin-28(n719) I*<br>Figure: 10B |
| Antibody | Monoclonal mouse anti-FLAG M2-Peroxidase (HRP) | Sigma-Aldrich | CAT#A8592 | Western Blot (1:1000) |
| Antibody | Monoclonal mouse anti-Actin clone C4 | Millipore | CAT#MAB1501 | Western Blot (1:10000) |

*Continued on next page*

*Continued*

| Reagent type (species) or resource | Designation | Source | Identifiers | Additional information |
|---|---|---|---|---|
| Antibody | Horseradish peroxidase-conjugated anti-mouse secondary antibody | GE Healthcare | CAT#NXA931 | Western Blot (1:7500) |
| Chemical compound, drug | Levamisol hydrochloride | Fluka Analytical | CAT#31742 | |
| Chemical compound, drug | Firefly D-Luciferin | p.j.k. | CAT#102111 | |
| Chemical compound, drug | 3-indoleacetic acid ('auxin') | Sigma-Aldrich | CAT#I2886 | |
| Commercial assay or kit | ImProm-II Reverse Transcription System | Promega | CAT#A3800 | |
| Commercial assay or kit | PowerUp SYBR Green Master Mix | Thermo Fisher Scientific | CAT#A25742 | |
| Commercial assay or kit | NucleoBond Xtra Midi | Macherey-Nagel | CAT#740410.50 | |
| Commercial assay or kit | NucleoBond Finalizer Plus | Macherey-Nagel | CAT#740520.20 | |
| Commercial assay or kit | Zymoclean Gel DNA Recovery Kit | Zymo Research | CAT#D4001 | |
| Recombinant DNA reagent | | | | Plasmid used in this study are listed in *Supplementary file 1A* |
| Sequence-based reagent | | | | Oligonucleotides used in this study are listed in *Supplementary file 1B* |

*These lines have been derived from a strain containing *him-5(e1490)* and *wIs54[scm::gfp] V*. We have not validated the presence of either mutation or transgene in the derived strain.

## *C. elegans*

Worm strains used in this study are listed in Key resources table. Bristol N2 was used as the wild-type strain. Animals were synchronized as described (*Aeschimann et al., 2017*) and, unless specified otherwise, grown on 2% NGM agar plates with *Escherichia coli* OP50 bacteria (*Stiernagle, 2006*). For microscopy, animals were mounted on a 2% (w/v) agarose pad and immobilized in 10 mM levamisol (Fluca Analytical, 31742).

## Generation of *lin-29b* and *lin-29a* mutant alleles using CRISPR-Cas9

Genome editing was performed as described (*Katic et al., 2015*) to obtain the following null mutant alleles using the sgRNAs listed in *Supplementary file 1B*. *lin-29(xe116)* is a 6724 bp deletion deletion of the putative *lin-29b* promoter region spanning the intron between exon 4 and exon 5 to disrupt *lin-29b* transcription. The deletion has the following flanking sequences: 5' atacggtttcagaattaaggagaca – *xe116* deletion – cctagatcaattgagctctaaagat 3'.

*lin-29(xe120)* is an insertion of an upstream ATG start codon, resulting in out-of-frame translation on the *lin-29b* mRNA. We introduced three point mutations in exon five to generate the following sequence: 5' ttcgaacaaaagcc(g→a)ga(c→t)gt(g→c) 3'. This introduces an AUG start codon upstream of the normal lin-29b AUG in an improved Kozak context to increase the efficiency of aberrant translation initiation. All three mutations are silent with respect to the *lin-29a* ORF.

*lin-29(xe200)* is a 1187 nt deletion spanning exons 2–4 that results in translation of a LIN-29a protein lacking the first 119 amino acids. 23 N-terminal amino acids remain upstream of the LIN-29b N-terminus. The deletion has the following flanking sequences: 5' ccaacttcttcaacgcaatg – 1187 nt deleted – ttttcacaatttggaagata 3'.

Injection of the same respective sets of sgRNAs into *lin-29(xe61[lin-29::gfp::3xflag])* generated *lin-29(xe61 xe114)*, *lin-29(xe61 xe121)*, *lin-29(xe61 xe133)*, which contain the respective mutations in the context of a GFP-3xFLAG-tagged LIN-29.

## Construction of plasmids for single-copy integrations

All final and intermediate plasmids as well as the primers used in these clonings are listed in *Supplementary file 1A*. The genomic regions containing coding exons (from the ATG start codon until the end of the 3'UTR) for *mab-10*, *lin-29a* and *lin-29b* were amplified by PCR using Phusion High-Fidelity DNA Polymerase (NEB, M0530S) from purified genomic wild-type *C. elegans* DNA. The *lin-29a* region was amplified in two fragments, the first spanning exons 2–4 (from ATG in exon 2), the second spanning exons 5–11 (including the 3'UTR). The large intron 4 was not amplified and exons 4 and 5 were fused into one exon when combining the two fragments during Gibson assembly. *lin-29a* 5'UTR was excluded as it is non-coding and confers LIN-41-mediated regulation (*Aeschimann et al., 2017*).

Using Gibson assembly, the PCR products were cloned into the BamHI site of pFA198. Plasmid pFA198 is a pENTR L1-L2 backbone plasmid that was constructed using Gateway cloning (BP reaction) to insert a 3xFLAG tag followed by a BamHI restriction site for subsequent Gibson assembly. Similarly, a FLAG-HA-degron tag followed by a BamHI restriction site was inserted into a pENTR L1-L2 backbone to obtain pFA218, a template for Gibson assembly reactions (*Aeschimann et al., 2019*). Plasmids with *mab-10*, *lin-29a* and *lin-29b* genomic regions cloned into pFA198 (i.e. pFA199, pFA206, pFA207) were only used as cloning intermediates in this study. From those plasmids, the *mab-10*, *lin-29a* and *lin-29b* genomic regions were re-amplified by PCR using Phusion High-Fidelity DNA Polymerase and inserted into BamHI-digested pFA218 using Gibson assembly. The resulting plasmids pFA219, pFA220 and pFA221 were then used for further Gateway cloning (LR reactions). In those LR reactions, transgenes were combined with the *col-10* promoter upstream (*Supplementary file 1A*) and an operon linker with *gfp-h2b::tbb-2*$_{3'UTR}$ (*Merritt et al., 2008*) downstream and cloned into the destination vector pCFJ150 (*Frøkjær-Jensen et al., 2008*). The resulting plasmids pFA238, pFA239 and pFA240 were used for injections.

## Construction of worm strains precociously expressing *lin-29* isoforms or *mab-10*

After constructing the plasmid for single copy integration, worm lines with integrated transgenes were obtained by single-copy integration into chromosome II (ttTi5605 locus) or chromosome V (oxTi365), using a protocol for injection with low DNA concentration (*Frøkjær-Jensen et al., 2012*). In order to perform injections while inducing auxin-mediated degradation of proteins expressed from the introduced transgenes, the Mos1 insertion strains EG6699 and EG8082 were crossed to worms expressing TIR1 from the *eft-3* promoter (IFM155) to generate strains HW2349 and HW2350, respectively. Following injections, these were grown on NGM plates supplemented with 1 mM auxin (Sigma-Aldrich, I2886). Injection mixes contained pCFJ601 (*Peft-3::transposase*) at 10 ng/ul, pGH8 (*Prab-3::mCherry*) at 10 ng/ul, pCFJ90 (*Pmyo-2::mCherry*) at 2.5 ng/ul, pCFJ104 (*Pmyo-3::mCherry*) at 5 ng/ul and the respective targeting vector at 50 ng/µl in water. Integrated transgenic lines were further crossed with worms expressing the *ajm-1::mCherry* marker.

## Seam cell and alae imaging and quantification

Arrested L1 larvae were grown at 25°C for 36–38 hr (late L4 stage) or 40–42 hr (young adult stage), with the developmental time assessed by staging of individual worms according to gonad length and vulva morphology. Fluorescent and Differential Interference Contrast (DIC) images were acquired with a Zeiss Axio Observer Z1 microscope using the AxioVision SE64 software and Zen 2 (blue edition). Selections of regions and processing of images was performed with Fiji (*Schindelin et al., 2012*).

Seam cell quantification was performed by counting all clearly visible fluorescent cells expressing an *scm::gfp* transgene (*Koh and Rothman, 2001*) of the upper lateral side in mounted worms.

Seam cell fusion quantification was performed by counting seam cell junctions visible through expression of an *ajm-1::mCherry* transgene (*Lehrbach et al., 2009*).

Alae quantification was performed by observing alae structures by DIC with a 100x objective. For worm lines throwing mnC1-balanced progeny, *Pmyo-2*::GFP-positive (i.e. balancer carrying) animals were excluded from imaging and quantifications. To score *let-7(xe150)* homozygous animals within a population of balanced animals, all *Pmyo-2::mCherry* expressing (i.e. balancer carrying) animals were excluded from the analysis. To score *lin-41(xe8)* homozygous animals within a population of

balanced *lin-41(xe8)/lin-41(bch28 xe70)* animals, all *Peft-3::gfp::h2b* expressing (i.e. balancer carrying) animals were excluded from the analysis.

## Luciferase assay

Luciferase assays were performed as described (*Meeuse et al., 2020*). Briefly, embryos were extracted from gravid adults using a bleaching solution. Single embryos were transferred into a well of a white, flat-bottom, 384-well plate (Berthold Technologies, 32505) by pipetting. Animals were left to develop in 90 µL S-Basal medium containing *E. coli* OP50 ($OD_{600}$ = 0.9) and 100 µM Firefly D-Luciferin (p.j.k., 102111). Plates were sealed with Breathe Easier sealing membrane (Diversified Biotech, BERM-2000). Luminescence was measured using a luminometer (Berthold Technologies, Centro XS3 LB 960) every 10 min for 0.5 s for 90 or 100 hr in a temperature controlled incubator set to 20°C. Luminescence data was analyzed using an automated algorithm to detect the hatch and the molts (*Meeuse et al., 2020*).

## Confocal imaging

Before acquiring images of representative worms, the GFP signals for at least 10 worms were observed to verify that they were comparable among different worms in each worm line and for each condition. For detection of endogenously tagged LIN-29 (HW1826, HW1882, HW1835) by confocal microscopy, animals were grown at 25°C. For confocal imaging of endogenously tagged LIN-29 (HW1826, HW1882, HW1835) and MAB-10 (HW2047) on RNAi conditions, synchronized arrested L1 stage larvae were grown for 20 or 22 hr at 25°C on RNAi-inducing plates with HT115 bacteria (*Ahringer, 2006*; *Timmons et al., 2001*) containing the L4440 RNAi vector either without insert ('mock RNAi') or with an insert targeting *lin-41* (*Fraser et al., 2000*) or *hbl-1* (*Kamath et al., 2003*). Worms were imaged on a Zeiss LSM 700 confocal microscope driven by Zen 2012 Software. Differential Interference Contrast (DIC) and fluorescent images were acquired with a 40x/1.3 oil immersion objective (1024 × 1024 pixels, pixel size 156 nm).

For confocal imaging of endogenously tagged LIN-29 in the wild-type and *lin-28(n719)* mutant background (HW1826, HW1882, HW1835, HW1924, HW1925, HW1926), worms were grown at 25°C for 20, 22, 24 and 22, 24, 26 hr, respectively (to accommodate a modest delay in development of *lin-28(n719)* mutants). For confocal imaging of endogenously tagged LIN-29a in the wild-type and *lin-29a(ΔN)* background (HW1826, HW2408), worms were grown at 25°C. In both cases, worms were imaged on an Axio Imager M2 (upright microscope) + Yokogawa CSU W1 Dual camera T2 spinning disk confocal scanning unit driven by Visiview 3.1.0.3. Differential Interference Contrast (DIC) and fluorescent images were acquired with a 40x/1.3 oil immersion objective (2048 × 2048 pixels, 16-bits). Using the Fiji software (*Schindelin et al., 2012*), images were processed after selecting representative regions. Worms of the same worm line were imaged and processed with identical settings.

## Testing for precocious seam cell fusion in worms precociously expressing *lin-29* isoforms or *mab-10*

All strains were grown on OP50-seeded NGM plates supplemented with 1 mM auxin. Gravid animals were bleached and eggs were incubated at room temperature in M9 for hatching overnight. To allow accumulation of LIN-29a, LIN-29b or MAB-10, the animals were not exposed to auxin anymore from this point onward. Synchronized L1 animals were plated and grown at 25°C. Seam cell fusion was scored by observation of the *ajm-1::mCherry* marker at 15 hr after plating synchronized L1 animals. Fluorescent and Differential Interference Contrast (DIC) images were acquired with a 40x/2.5 oil immersion objective with a Zeiss Axio Observer Z1 microscope using the AxioVision SE64 software and Zen 2 (blue edition). Selections of regions and processing of images was performed with Fiji (*Schindelin et al., 2012*).

## RT-qPCR

Reverse transcription was performed with the ImpromII Reverse Transcription System (Promega; A3800), according to the manufacturer's protocol, with 100 ng RNA (for *lin-41* and *hbl-1* samples) or 80 ng RNA (for *lin-28(n719)* samples) and random primers (Promega; C1181). Using Power Up SYBR Green Master Mix (Thermo Fisher Scientific;A25742), qPCR was performed on a Light Cycler 480II 384 (Roche) with the primers listed in the *Supplementary file 1B*. For comparing mRNA levels of *lin-*

*29b* of worms grown on mock, *lin-41* or *hbl-1* RNAi bacteria, -ΔΔCt were calculated using *act-1* as an internal control mRNA and the mock RNAi condition (first time point) as calibrator. For comparing mRNA levels of *lin-29a* and *lin-29b*, of wild-type or *lin-28(n719)* worms, -ΔΔCt were calculated using *act-1* as an internal control mRNA and the wild-type condition (first time point) as calibrator.

## Western blotting

Animals were grown for 20 hr at 25°C on RNAi-inducing plates, as described above for confocal imaging, or on NA22 plates (*Evans, 2006*). Lysates were made by boiling (5 min, 95°C) and sonication in SDS lysis buffer (63 mM Tris-HCl (pH 6.8), 5 mM DTT, 2% SDS, 5% sucrose) and cleared by centrifugation, before separating proteins by SDS-PAGE (loading: 50 μg protein extract per well) and transferring them to PVDF membranes by semi-dry blotting. The following antibodies were used: Monoclonal mouse anti-FLAG M2-Peroxidase (HRP) (Sigma-Aldrich; A8592, dilution: 1:1000). Monoclonal mouse anti-Actin clone C4 (Millipore; MAB1501, dilution 1:10000). Detection was performed with a horseradish peroxidase-conjugated secondary anti-mouse antibody (NXA931), ECL Western Blotting Detection Reagents and an ImageQuant LAS 4000 chemiluminescence imager (all from GE Healthcare).

## Quantification and statistical analysis

All values of n indicate numbers of animals. No statistical tests were performed.

## Acknowledgements

We thank Milou Meeuse for the gift of HW1949 and help with the luciferase assay, Jana Kracmarova for the gift of *let-7(xe150)* and of RNA samples used for the RT-qPCR experiment (*Figure 10B,C*), Iskra Katic for the gift of IFM155, and Foivos Gypas for computational analysis. Some strains were provided by the Caenorhabditis Genetics Center, which is funded by the NIH Office of Research Infrastructure Programs (P40 OD010440). We thank Iskra Katic, Jana Kracmarova for a critical reading of the manuscript. We are particularly grateful to Benjamin Towbin for extensive comments, suggestions and discussions. We thank Lan Xu and Iskra Katic for technical support. This project was supported through funding from the Swiss National Science Foundation (#31003A_163447 and #310030_188487) and Friedrich Miescher Institute for Biomedical Research core funding through the Novartis Research Foundation (to HG).

---

# Additional information

### Competing interests

Florian Aeschimann: The author is now affiliated with CSL Behring, Research, CSL Biologics Research Center, however all work was conducted when affiliated with Friedrich Miescher Institute for Biomedical Research and University of Basel. The other authors declare that no competing interests exist.

### Funding

| Funder | Grant reference number | Author |
|---|---|---|
| Schweizerischer Nationalfonds zur Förderung der Wissenschaftlichen Forschung | 31003A_163447 | Helge Großhans |
| Schweizerischer Nationalfonds zur Förderung der Wissenschaftlichen Forschung | 310030_188487 | Helge Großhans |
| National Institutes of Health | NIH Office of Research Infrastructure Programs P40 OD010440 | Helge Großhans |
| Friedrich Miescher Institute for Biomedical Research | Core funding through the Novartis Research Foundation | Helge Großhans |

The funders had no role in study design, data collection and interpretation, or the decision to submit the work for publication.

## Author contributions
Chiara Azzi, Conceptualization, Formal analysis, Investigation, Writing - original draft, Writing - review and editing, designed, performed, and analyzed confocal microscopy, phenotypic analysis, luciferase assay, RT-qPCR and western blot experiments; created transgenic worm lines; Florian Aeschimann, Conceptualization, Formal analysis, Investigation, Writing - review and editing, designed, performed, and analyzed confocal microscopy, phenotypic analysis and western blot experiments; created transgenic worm lines; Anca Neagu, Resources, Formal analysis, Investigation, performed and analyzed the precocious expression experiment and created transgenic worm lines; Helge Großhans, Conceptualization, Supervision, Funding acquisition, Investigation, Writing - original draft, Project administration, Writing - review and editing

## Author ORCIDs
Chiara Azzi (iD) https://orcid.org/0000-0002-2240-4618
Florian Aeschimann (iD) https://orcid.org/0000-0001-5213-034X
Anca Neagu (iD) https://orcid.org/0000-0002-0643-5238
Helge Großhans (iD) https://orcid.org/0000-0002-8169-6905

## Decision letter and Author response
Decision letter https://doi.org/10.7554/eLife.53387.sa1
Author response https://doi.org/10.7554/eLife.53387.sa2

# Additional files

## Supplementary files
• Supplementary file 1. (**A**) Plasmids used in this study. (**B**) Oligonucleotides and sgRNAs used in this study.
• Transparent reporting form

## Data availability
All data generated or analysed during this study are included in the manuscript and supporting files. Source data files have been provided for Figures 1, 3–6 and Figure 6—figure supplement 2.

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
