## [Decision Letter]

**Acceptance summary:**

This paper reports that two isoforms of the transcription factor LIN-29 exhibit partially overlapping but distinct roles in the heterochronic pathway. These results provide important new insights into the architecture of the heterochronic pathway and on the relationships between *lin-28, let-7*, and *hbl-1* in regulating the juvenile-to-adult transition.

**Decision letter after peer review:**

Thank you for submitting your article "A branched heterochronic pathway directs juvenile-to-adult transition through two LIN-29 isoforms" for consideration by *eLife*. Your article has been reviewed by two peer reviewers, and the evaluation has been overseen by a Reviewing Editor and Marianne Bronner as the Senior Editor. The following individuals involved in review of your submission have agreed to reveal their identity: Douglas Portman (Reviewer #1); Frank Slack (Reviewer #2).

The reviewers have discussed the reviews with one another and the Reviewing Editor has drafted this decision to help you prepare a revised submission.

Essential revisions:

1) Authors propose that fused seam cells can "un-fuse" and undergo an additional round of mitosis in *mab-10(0) lin-29a(∆)* mutants. This is a very plausible explanation for the phenotype shown in Figure 3B, but were lineage experiments carried out to determine that this actually happened? If not, the language here should be softened just a bit.

2) What is the asterisk in the lower-right panel of Figure 7C?

3) Figure 9A: *lin-29b*-expressing hyp7 nuclei are clearly visible in *hbl-1*(RNAi) (panel A) but are not labeled with arrowheads.

4) The authors find that *lin-28* mutants, the spatial expression of *lin-29a/b* is a bit different than in *lin-41* and *hbl-1* mutants. That is, *lin-29a* appears mostly in the seam (whereas it's only in hyp7 in *lin-41*) and *lin-29b* appears only in the seam, whereas it's also in hyp7 in *hbl-1*). These unexpected results seem worth some speculation.

5) Figure 4 – is "patches of alae" a more severe phenotype than "weak alae"? or just different?

6) It would be useful for the authors to incorporate *lin-46* into the Discussion. The Ambros lab has two recent papers (in Development and bioRxiv) showing that *lin-28* acts through *lin-46* to regulate *hbl-1*, but the current study does not mention *lin-46*. While additional experiments are noit necessary to tie these stories together, it seems important for this paper to speculate about the relationship of *lin-46* to the mechanisms they describe here.

7) The authors' efforts to reconcile their results with previous data are commendable and valuable. However, Discussion paragraph five, the scenario they describe for *lin-41* mutants would seem to make sense only if *lin-29a* can promote seam fusion non-autonomously.

8) The authors should show the quantified data related to Figure 3B (yA stage)

9) In Figure 2B LIN-29b protein levels in *lin-29a(xe40)* mutant animals seem to be higher and different than wild type animals. Explanation?

10) In Figure 9 (B) The LIN-29 b protein levels in *lin-41* RNAi (row 3) worms seem to be higher than mock RNAi worms (row 2). Explanation?

11) How do you explain the LIN-29b protein levels in *lin-41* RNAi (row 7) that are higher than mock (row 6)? As the control protein ACT-1 levels are not the same in these two groups.

12) In Figure 6—figure supplement 2 (C): The LIN-29 b protein levels in *lin-41* RNAi seem to be higher than mock RNAi worms. How do you explain these observations? Is there interaction between the two *lin-29* isoforms?

13) Do Pcol-10::*lin-29a* and Pcol-10::lin-29b have other abnormal phenotypes without seam cell fusion?

14) Do *lin-29b* mRNA levels change in *hbl-1* RNAi and *lin-41* RNAi worms?

How about *lin-29a* mRNA and *lin-29b* mRNA levels in *lin-28* mutants?

The authors should discuss these results in the Discussion section.

15) In Figure 9 A and Figure 10, LIN-29 b protein levels couldn't be seen in the epidermal cells in Mock RNAi worms and wild type worms, respectively. However we can see the LIN-29 b protein levels in the wild type animals in Figure 7 B that seem to be of the same (L3) stage. How do the authors explain this?

In addition, the following point was raised. We think that this is an important point that ideally, should be tested experimentally. However, if this is not possible in this timeframe, please address this textually and modify the text accordingly.

A significant concern has to do with cell-type specificity. Repeatedly the authors claim that the difference in the two LIN-29 isoforms stems from their spatio-temporal regulation, rather than their sequence. The sequence difference issue is very carefully addressed, as is temporal regulation, but relatively little attention is paid to the spatial (cell type) regulation. In order to better substantiate the authors' claims, we would find it worthwhile for the authors to more carefully dissect the role of LIN-29 in the seam cells vs. the hypodermis. In particular, it seems important to better understand the functional significance of the observations that *lin-29b* is expressed in seam cells by mid-L3 while *lin-29a* is not expressed there until mid-L4, and that both *lin-29a and lin-29b* are expressed in hyp7 by mid-L4. Presumably this is related to the particular importance of *lin-29b* for cell cycle exit and seam cell fusion, but specific manipulation of *lin-29* in the seam vs. hyp7 would be necessary to understand this.

---

## [Author Response]

Essential revisions:1) Authors propose that fused seam cells can "un-fuse" and undergo an additional round of mitosis in mab-10(0) lin-29a(∆) mutants. This is a very plausible explanation for the phenotype shown in Figure 3B, but were lineage experiments carried out to determine that this actually happened? If not, the language here should be softened just a bit.

We appreciate the reviewers’ point. To further clarify why we think this is the most likely explanation, we have now emphasized in the text that *mab-10(0);lin-29a(∆)* mutant L4 stage animals have completely fused seam cells, and that the phenotype is 100% penetrant. Hence, “un-fusion” of the seam cells during additional nuclear divisions is a parsimonious explanation of the divergent adult phenotype. Nonetheless, we agree that without lineaging, other, more complex scenarios, might not be ruled out and softened the language of this paragraph as requested.

2) What is the asterisk in the lower-right panel of Figure 7C?

We apologize for having omitted this piece of information in the original manuscript. In the lower-right panel of the original Figure 7C, LIN-29b is detected in the posterior daughter cells (arrowheads) and the anterior daughter cells (asterisks) that will fuse with hyp7. We have now replaced asterisks by arrows to be consistent with the other panels, where we didn’t distinguish between posterior and anterior daughter cells.

3) Figure 9A: lin-29b-expressing hyp7 nuclei are clearly visible in hbl-1(RNAi) (panel A) but are not labeled with arrowheads.

We apologize for the omission. We have now labeled the hyp7 nuclei in Figure 9A with arrowheads and mention this observation in the text.

4) The authors find that lin-28 mutants, the spatial expression of lin-29a/b is a bit different than in lin-41 and hbl-1 mutants. That is, lin-29a appears mostly in the seam (whereas it's only in hyp7 in lin-41) and lin-29b appears only in the seam, whereas it's also in hyp7 in hbl-1). These unexpected results seem worth some speculation.

We agree with the reviewers on this pertinent point and decided to investigate it more thoroughly by collecting data for additional time points to understand the dynamics of regulation. Specifically, we looked at the *lin-29a/b* tagged lines on *lin-41* RNAi at 20 and 22 hours of development and at 22, 24 and 26h in the *lin-28*(n719) mutant background (*lin-28* mutants are delayed in development by two hours). As shown in the new Figure 9—figure supplement 1B, LIN-29a is indeed undetectable at 20h in seam cells of *lin-41* RNAi animals, as we reported initially, but it becomes detectable (albeit more weakly than in hyp7) at 22h. Conversely, and as shown in the new Figure 10—figure supplement 1, in *lin-28*(n719) animals, LIN-29a accumulates precociously not only in seam cells but also in hyp7. Initially, accumulation is much stronger in seam than in *hyp7*, but increases considerably in hyp7 over time. We cannot exclude that some of these differences are explained by differences in the extent and kinetics of LIN-41 depletion (e.g., *lin-28* mutation acts indirectly, by increasing let-7 levels, which may not recapitulate the effects of earlier, chronic depletion of LIN-41 by RNAi). However, it appears possible, and we comment on it in the manuscript, that LIN-28 further affects an additional, seam-specific mechanism of *lin-29a* regulation that is independent of LIN-41.

5) Figure 4 – is "patches of alae" a more severe phenotype than "weak alae"? or just different?

Based on the available data, we are reluctant to distinguish between a “different” and a “more severe” phenotype. Visually, they clearly look very different, and they are associated with different mutations, namely *lin-29a*(∆) for weak alae (which in slightly older adults are not patchy, but cover the whole body) and *lin-29b*(∆) for patches of alae (which remain patches even in older animals).

6) It would be useful for the authors to incorporate lin-46 into the Discussion. The Ambros lab has two recent papers (in Development and bioRxiv) showing that lin-28 acts through lin-46 to regulate hbl-1, but the current study does not mention lin-46. While additional experiments are noit necessary to tie these stories together, it seems important for this paper to speculate about the relationship of lin-46 to the mechanisms they describe here.

We thank the reviewers for this suggestion. We are now discussing how *lin-46* might fit into the picture in the text and Figure 11.

7) The authors' efforts to reconcile their results with previous data are commendable and valuable. However, Discussion paragraph five, the scenario they describe for lin-41 mutants would seem to make sense only if lin-29a can promote seam fusion non-autonomously.

We agree that this is a reasonable criticism considering the previous data. However, the new Figure 9—figure supplement 1B now reveals that LIN-41 depletion induces LIN-29a accumulation not only in hyp7 but also in the seam (just slightly later). Hence, there is no need to invoke a tissue non-autonomous function.

8) The authors should show the quantified data related to Figure 3B (yA stage)

We included the data requested by the reviewers in Figure 3—source data 2.

9) In Figure 2B LIN-29b protein levels in lin-29a(xe40) mutant animals seem to be higher and different than wild type animals. Explanation?

LIN-29 levels are highly stage dependent and we can only approximate LIN-29, in this case LIN-29b, levels by Western blot. As can be seen also from the replicates in Author response image 1, there is no consistent trend for elevated LIN-29b in lin-29a(xe40) animals. What we do observe though is a trend towards higher levels of LIN-29a in the N-terminally tagged line vs. the line in which both a and b isoform are C-terminally tagged. (It is indeed possible that the reviewers misread the figure labels and were referring to this result – we have improved the label of this figure to address this possibility.) If we were to speculate, it seems possible that the C-terminal tag somewhat destabilizes the protein, consistent with our observation that this protein does not appear to be fully functional (i.e., animals carrying the C-terminal tag on *lin-29a* and not expressing *lin-29b* are somewhat sicker than the isogenic strain lacking the tag.) Since the experiments involve differential analysis of isogenic strains, this should not affect our conclusions.

**Author response image 1. respfig1:** Replicate Western blot experiments.

10) In Figure 9 (B) The LIN-29 b protein levels in lin-41 RNAi (row 3) worms seem to be higher than mock RNAi worms (row 2). Explanation?11) How do you explain the LIN-29b protein levels in lin-41 RNAi (row 7) that are higher than mock (row 6)? As the control protein ACT-1 levels are not the same in these two groups.12) In Figure 6—figure supplement 2 (C): The LIN-29 b protein levels in lin-41 RNAi seem to be higher than mock RNAi worms. How do you explain these observations? Is there interaction between the two lin-29 isoforms?

Comments 10–12 all make the same point, i.e., that manipulations of lin-41 might cause elevated LIN-29b levels. We don’t know if this effect is real. We occasionally observed what appeared to be an increase of LIN-29b on *lin-41* RNAi condition on Western Blots (Figure 9A, Figure 6—figure supplement 2C, Aeschimann, 2017 Figure 4E and S3D) but in every case, this increase appeared minor. More importantly, we were unable to observe it when inspecting animals under the microscope. Hence, we are reluctant to draw any conclusions. However, we do point out that *lin-29* RNA levels oscillate (Aeschimann et al., 2017). As the amplitude is small, small variations in strain growth are unlikely to affect outcomes when the effects are black and white (e.g*., lin-29b* level changes in response to HBL-1 depletion), but it may explain the more modest effects noticed by the reviewers.

13) Do Pcol-10::lin-29a and Pcol-10::lin-29b have other abnormal phenotypes without seam cell fusion?

We observed other morphological phenotypes by *col-10*-driven expression of either *lin-29* isoform: animals were small, and had no, or very few progeny. To investigate the sterility phenotype, we quantified the number of hatched eggs (from P0s that were singled out at the L4 stage to OP50 plates without auxin and allowed to lay eggs for 1 day), as well as the presence of offspring in the next generation. We did not observe any hatching defects, but fertility was highly impaired (data shown in the Author response tables 1 and 2). Additionally, we observed that precocious expression of either *lin-29* isoform caused retarded gonad development and abnormal vulval formation with delayed invagination of the vulval epithelium (Author response image 2). Vulval abnormalities, small size and reduced fertility upon misexpression of *lin-29a* were recently also reported by Abete-Luzi et al., 2020. Since we have not explored the basis of these phenotypes, we have not included these data in the revised manuscript but present them in Author response tables 1 and 2 and Author response image 2.

**Author response table 1. resptable1:** 

Line	Hatched	Unhatched	Total
*Pcol-10::lin-29a*	96	4	100
*Pcol-10::lin-29b*	95	5	100
*Pcol-10::mab-10*	95	5	100
*N2*	95	5	100

**Author response table 2. resptable2:** 

Line	Fertile^1^	Sterile	Censored^2^	Total
*Pcol-10::lin-29a*	8	15	7	30
*Pcol-10::lin-29b*	16	5	9	30
*Pcol-10::mab-10*	24	2	4	30
*N2*	30	0	0	30

^1^Even fertile *lin-29* and *mab-10* lines had greatly reduced brood sizes, varying between 1 and 15 progeny.

^2^Missing animals, and those that had dried-out on the walls of the plate or burst early.

**Author response image 2. respfig2:** Vulval phenotypes upon precocious lin-29 expression.

14) Do lin-29b mRNA levels change in hbl-1 RNAi and lin-41 RNAi worms?How about lin-29a mRNA and lin-29b mRNA levels in lin-28 mutants?The authors should discuss these results in the Discussion section.

To address these questions, we performed time course experiments followed by RT-qPCR. Specifically, to investigate *lin-29b* mRNA levels in *hbl-1* RNAi and *lin-41* RNAi conditions, we collected worms from 6 to 20 hours of development every two hours. As shown in the new Figure 9—figure supplement 2, lin-29b mRNA is indeed precociously upregulated upon *hbl-1* RNAi, but not lin-41 RNAi. (Note that levels converge again at 22h, when HBL-1 is also down-regulated in the wild-type condition.) Regarding the *lin-29a* and *lin-29b* mRNA levels in *lin-28* mutants, the new Figure 10B-C reveals that this mutation causes upregulation of the *lin-29b* but not the *lin-29a* transcript, consistent with functions mainly through *hbl-1* and *lin-41*, respectively.

15) In Figure 9 A and Figure 10, LIN-29 b protein levels couldn't be seen in the epidermal cells in Mock RNAi worms and wild type worms, respectively. However we can see the LIN-29 b protein levels in the wild type animals in Figure 7 B that seem to be of the same (L3) stage. How do the authors explain this?

The stage of the worm in the two experiments is different, within the L3 larval stage. In fact, the endogenous LIN-29b protein levels in the wild-type animals start to accumulate in the hypodermis in late L3 worms, as shown and indicated in Figure 7B. Instead, we performed the RNAi experiment earlier, at 20 hours of development, as shown in Figure 9A and 10. We chose on purpose a time in which the LIN-29a and LIN-29b proteins were not detectable in order to underline the precocious expression of the *lin-29* isoforms in *lin-41* and *hbl-1* RNAi condition; this is now emphasized in the text.

In addition, the following point was raised. We think that this is an important point that ideally, should be tested experimentally. However, if this is not possible in this timeframe, please address this textually and modify the text accordingly.A significant concern has to do with cell-type specificity. Repeatedly the authors claim that the difference in the two LIN-29 isoforms stems from their spatio-temporal regulation, rather than their sequence. The sequence difference issue is very carefully addressed, as is temporal regulation, but relatively little attention is paid to the spatial (cell type) regulation. In order to better substantiate the authors' claims, we would find it worthwhile for the authors to more carefully dissect the role of LIN-29 in the seam cells vs. the hypodermis. In particular, it seems important to better understand the functional significance of the observations that lin-29b is expressed in seam cells by mid-L3 while lin-29a is not expressed there until mid-L4, and that both lin-29a and lin-29b are expressed in hyp7 by mid-L4. Presumably this is related to the particular importance of lin-29b for cell cycle exit and seam cell fusion, but specific manipulation of lin-29 in the seam vs. hyp7 would be necessary to understand this.

In order to address the reviewers’ concern, we tried to substitute, in the lines HW2505 and HW2604 that express *lin-29a* or *lin-29b* from a single-copy transgene, the “pan-epidermal” col-10 promoter with other promoters. Specifically, we used the seam cell-specific promoter bro-1 and the hyp7-specific promoter elt-3. We then wanted to examine precocious seam cell fusion events. In the given time, we were able to generate the lines expressing *lin-29a* or *lin-29b* under the control of the bro-1, but not under the control of the elt-3 promoter. We found that bro-1-driven expression of either isoform induced precocious seam cell fusion. Specifically, we observed that at 13 hours after plating, at the L1 molt, all animals had at least partially fused seam cells. Obviously, this experiment is now lacking an important component, that is, we can conclude that expression of *lin-29* in seam cells is sufficient for fusion, but we do not know whether it is necessary. Hence, we decided against including this experiment in the manuscript and instead, and as recommended, pointed out this limitation of the study in the text.

**Author response image 3. respfig3:** Seam cell fusion upon precocious lin-29 expression in only the seam.